# FREQUENCY-CONDITIONED DIFFUSION MODELS FOR TIME SERIES GENERATION

## ABSTRACT

Time series data, commonly used in fields like climate studies, finance, and healthcare, usually faces challenges such as missing data and privacy concerns. Recently, diffusion models have emerged as effective tools for generating high-quality data, but applying them to time series is still difficult, especially for capturing long-range dependencies and complex information. In this paper, we introduce a new diffusion model that uses frequency domain information to improve time series data generation. In particular, we apply Fourier analysis to adaptively separate low-frequency global trends from high-frequency details, which helps the model better understand important patterns during the denoising process. Finally, our approach uses a specialized frequency encoder to integrate this information, enhancing the model's ability to capture both global and local features. Through exhaustive experiments on various public datasets, our model shows an impressive performance in generating time series data for diverse tasks like forecasting and imputation, outperforming existing methods in accuracy and flexibility.

## 1 INTRODUCTION

Time series data, which records observations over time, is widely used in various real-world fields such as climate studies (Mudelsee, 2010), finance (Koa et al., 2023), and healthcare (Jeong et al., 2024). However, the process of collecting such data often faces significant challenges, including the need to simulate scenarios that did not occur during data collection (Koa et al., 2023) and concerns over personal privacy (Yoon et al., 2020). To address these issues, there has been a surge in research focused on synthesizing time series data in recent years. Generative models such as generative adversarial networks (GANs), variational autoencoders (VAEs), and their variants have achieved notable success in this area (Ang et al., 2023). With this success, there have been numerous extensions to generative models, including decomposing time series data elements to improve interpretability (Desai et al., 2021), and developing models for imputation (Tashiro et al., 2021) and forecasting (Rasul et al., 2021).

Recently, in the field of computer vision, the dominant approach to generative modeling has shifted towards score-based diffusion models (Ho et al., 2020). These models offer the benefit of generating high-quality data while addressing the limitations of traditional generative models like GANs and VAEs. Building on the success of diffusion-based modeling in computer vision, there is a growing interest in applying those techniques to time series data analysis (Ang et al., 2023). Initially, research focused on task-specific generation like imputation (Tashiro et al., 2021) and prediction (Yan et al., 2021). It has since expanded to explore more general approaches to time series data generation (Lim et al., 2023; Yuan & Qiao, 2024).

Despite the success of diffusion-based time series generation, the existing methods encounter several critical challenges. First, autoregressive models are limited in long-range performance due to the accumulation of errors, and their inference speed is slow because data is generated iteratively (Lim et al., 2023). In this regard, recent research on the diffusion model in time series modeling is more directed to non-autoregressive methods to address the issue of error accumulation. However, those approaches make it nearly impossible to capture all time-related information, particularly in long-range sequences, making it a highly challenging problem. To handle this, time series data is typically divided into smaller windows for learning (Yuan & Qiao, 2024), but this method has trouble effectively modeling gradually changing trends as depicted in Figure 1. Additionally, unconditional

generative models struggle to effectively model the temporal-spatial-spectral information inherent in multivariate time series data, particularly during the denoising process, where capturing this complex information proves to be challenging.

Fourier analysis provides a powerful way to extract rich information by transforming time-domain data into the frequency domain, offering a natural solution to challenges in long-distance sequences, as it captures both global and local characteristics in time series more effectively. Specifically, global trends are concentrated in the low-frequency domain, while semantic details, such as spikes, are isolated in the high-frequency domain. These distinct characteristics have significantly contributed to the success of generative models, as they allow for better representation and modeling of complex data patterns (Lee et al., 2023; Crabbé et al., 2024). Notably, while information is evenly distributed across time in the time domain, it becomes concentrated in the low-frequency region in the frequency domain. This concentration simplifies the model's task of learning scores during the denoising process,

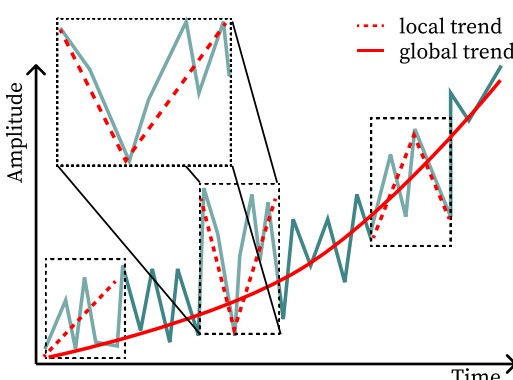

Figure 1: Illustration of the gap that occurs between the trend observed within a specific interval and the overarching trend present in the data.

enhancing overall performance (Crabbé et al., 2024). However, relying solely on modeling in the frequency domain is not ideal. While time-domain signals inherently contain frequency information, the frequency domain loses temporal information after transformation. This weakens the model's inductive bias for learning predictive ability, a crucial aspect of time series data. Therefore, balancing both time and frequency domains is essential to fully capture and leverage the diverse characteristics of time series data.

In this paper, we propose a novel diffusion model that incorporates time-domain information while leveraging rich frequency-domain information to address the aforementioned limitations. By utilizing the distinct characteristics of low and high frequencies, we simplify the complex process of decomposing temporal components. This enables us to condition the model on frequency information during the denoising process, resulting in the generation of higher-quality samples. Unlike the previous approach (Lee et al., 2023), where frequency information was arbitrarily divided, we dissect the frequency components based on the amount of information present in the spectral density. This dissected frequency information is then cross-attended within the denoising structure through a frequency encoder, allowing the model to more effectively capture and utilize both global and local features during generation. To demonstrate the validity and robustness of our proposed method, we conducted exhaustive experiments, achieving promising results across various public datasets. Moreover, our method excels in task-specific generation, such as prediction and imputation, where each frequency component plays a critical role, showcasing the model's versatility across diverse time series generation challenges.

In summary, the major contributions of our work are as follows:

- We propose a novel time series generation diffusion model that operates in the time domain while utilizing frequency as prior knowledge. This approach enables the model to effectively capture and represent the characteristics of the data by combining time-domain and frequency-domain information during the diffusion learning process.

- We introduce a module that adaptively dissects frequency information based on the power spectrum, leading the model to effectively represent temporal aspects in the data. This facilitates high-quality synthesis, which more accurately reflects the inherent trends and patterns of the dataset.

- We demonstrate the effectiveness of our proposed method for generating time series data across various public datasets, showing that it achieves superior performance not only in long-range data generation but also in imputation and forecasting tasks.

## 2 RELATED WORKS

### 2.1 GENERATIVE MODELS IN TIME SERIES

Deep generative models designed for time series data analysis, especially GAN-based models, have evolved to integrate temporal dynamics and spatial features into their architectures effectively. In C-RNN-GAN (Mogren, 2016), the authors introduced a method that integrates GAN with a recurrent model, marking the first approach to applying GANs to sequential data. Subsequently, TimeGAN (Yoon et al., 2019) introduced a generative model that combines an embedding function and a supervised loss with the original GAN architecture to better control conditional temporal dynamics. GT-GAN (Jeon et al., 2022) addressed the irregularity of time series data using a neural ODE-based approach, and PSA-GAN (Jeha et al., 2022) was developed to progressively enhance GAN performance through self-attention mechanisms. Meanwhile, VAE-based generative models have also developed rapidly, offering key advantages such as fast sampling speed and high diversity in the generated data. TimeVAE (Desai et al., 2021) aimed to enhance the interpretability of time series data by generating it through temporal components. In TimeVQVAE (Lee et al., 2023), time series data is synthesized using vector quantization techniques by separating the data into low-frequency and high-frequency components within the frequency domain. Furthermore, to generate medical time series data that accounts for causality, CR-VAE (Li et al., 2023) introduced a model that integrates causal mechanisms into the recurrent VAE architecture.

Consequently, various time series data synthesis models based on GANs and VAEs have been developed. However, these models face inherent limitations—GANs struggle with adversarial loss training and mode collapse, while VAEs often produce blurry samples and face challenges in optimizing the KL divergence. As a result, achieving both high-quality samples and sufficient diversity remains a significant challenge.

### 2.2 DIFFUSION MODELS IN TIME SERIES

Recently, diffusion-based models have become a central focus of research, surpassing other generative models by providing a methodology that successfully achieves both high quality and diversity in generated data. Significant progress is also being made in applying these models to time series data. CSDI (Tashiro et al., 2021) introduced a diffusion model that imputes missing values by conditioning on observed data, utilizing a self-supervised learning approach. Additionally, for prediction tasks, TimeGrad Rasul et al. (2021), based on DDPM, and ScoreGrad (Yan et al., 2021), based on SDE, were proposed. These models integrate the temporal feature modeling of RNNs with diffusion processes, enhancing the predictive performance of time series models. While these methods initially struggled with error accumulation in long-range predictions, recent advances have introduced non-autoregressive approaches, such as TimeDiff (Shen & Kwok, 2023), which incorporates future information, and LDT (Feng et al., 2024), which models in latent space. In addition to task-specific models like imputation and prediction, recent research has increasingly focused on diffusion-based models for more general tasks that aim to learn data distribution. In TSGM (Lim et al., 2023), a generative model is developed by combining an RNN-based autoencoder with a conditional diffusion model. In DiffTime (Coletta et al., 2024), the authors addressed the challenge of generating data under constraints like trends and fixed values and proposed a diffusion model specifically designed to operate within these constraints. As another approach, the Diffusion-TS (Yuan & Qiao, 2024) was developed to generate data by disentangling temporal components such as trends and seasonalities. In Time Weaver (Narasimhan et al., 2024), the authors improved performance by combining heterogeneous paired metadata. On the other hand, recent research (Crabbé et al., 2024) has shown that the frequency domain can be a more effective inductive bias for diffusion modeling compared to the time domain, based on both theoretical and experimental analyses. Still, the frequency domain alone lacks the ability to capture temporal information. To address this, we integrate frequency domain information with time domain modeling in a multi-view approach, leading to improved performance on real-world datasets.

## 3 BACKGROUNDS

### 3.1 DENOISING DIFFUSION PROBABILISTIC MODELS

Diffusion-based models are a type of generative model known for producing high-quality and diverse outputs. Among these, the Denoising Diffusion Probabilistic Model (DDPM) (Ho et al., 2020) is particularly well-known, featuring a forward process that gradually adds noise to the data and a reverse process that reconstructs the data by removing the noise.

During the forward process, the input value $\mathbf{x}_0 \sim q(\mathbf{x})$ is gradually noised into standard Gaussian noise $\mathbf{x}_T \sim \mathcal{N}(0, \mathbf{I})$ by incrementally adding noise at each diffusion step $t$:

$$q(\mathbf{x}_t|\mathbf{x}_{t-1}) = \mathcal{N}(\mathbf{x}_t; \sqrt{1 - \beta_t}\mathbf{x}_{t-1}, \beta_t\mathbf{I}), \quad t = 1, \cdots, T \tag{1}$$

where $\beta_t \in (0, 1)$.

The reverse process is defined as a Markovian process, which gradually denoises samples through reverse transitions:

$$p_\theta(\mathbf{x}_{t-1}|\mathbf{x}_t) = \mathcal{N}(\mathbf{x}_{t-1}; \mu_\theta(\mathbf{x}_t, t), \Sigma_\theta(\mathbf{x}_t, t)) \tag{2}$$

where $\mu_\theta(\cdot)$ is defined by a neural network and $\Sigma_\theta(\cdot)$ is typically fixed as $\sigma_t^2\mathbf{I}$.

The reverse process is handled by a surrogate approximation that parameterizes $\mu_\theta(\mathbf{x}_t, t)$ at each diffusion step $t$. Hence, the denoising model parameters $\theta$ are optimized by minimizing:

$$\mathcal{L}(\mathbf{x}_0) = \sum_{t=1}^{T} \mathbb{E}_{q(\mathbf{x}_t|\mathbf{x}_0)}||\mu(\mathbf{x}_t, \mathbf{x}_0) - \mu_\theta(\mathbf{x}_t, t)|| \tag{3}$$

where $\mu(\mathbf{x}_t, \mathbf{x}_0)$ denotes the mean of posterior $q(\mathbf{x}_{t-1}|\mathbf{x}_0, \mathbf{x}_t)$.

### 3.2 FOURIER ANLNYSIS

Fourier analysis is a mathematical method that transforms signals from the time domain to the frequency domain. It shows remarkable performance in applications such as signal processing, data compression, and machine learning by breaking down signals into their frequency components, which simplifies their analysis and manipulation (Körner, 2022).

**Signal energy.** Signal energy is a crucial characteristic in signal analysis, representing the total energy contained within a signal. Let $\mathbf{x}$ be the signal and $\mathbf{x}'$ be the transformed signal into the frequency domain. It is determined by the amplitude and duration of the signal and can be calculated using the squared Frobenius norm $||\mathbf{x}||^2$. According to Parseval's theorem, the signal energy remains the same whether calculated in the time domain or the frequency domain, $||\mathbf{x}||^2 = ||\mathbf{x}'||^2$.

When training a diffusion model, relying solely on the energy density, which is evenly distributed over time, may not be sufficient (Crabbé et al., 2024). That is, there is a limit to time domain information. To address this, we condition the model by separating the frequency components into low and high frequencies based on spectral energy density. This separation allows the model to focus on different aspects of the signal, enhancing its ability to capture both global and local features during the learning process.

## 4 FREQUENCY-CONDITIONED DIFFUSION MODELS FOR TIME SERIES

Consider a multivariate time series $\mathbf{x} \in \mathbb{R}^{L \times F}$ with time length $L$ and dimension $F$. As discussed in Section 3, the time series $\mathbf{x}$ is transformed into the frequency domain $\mathbf{x}' = \mathcal{F}(\mathbf{x})$ using fourier transformation (Elliott & Rao, 1982). Given a dataset $D_X = \{(\mathbf{x}_i, \mathbf{x}_i')\}_{i=1}$, our objective is to develop a conditional generation model $G$. During the reverse process of the training phase, these transformed frequency information acts as conditions to facilitate smooth denoising. Notably, in the inference step, frequency information is directly extracted from the dataset and utilized.

As aforementioned, time series data show complicated patterns in situations in the real world. Existing diffusion-based time series generation models have predominantly focused on time-domain

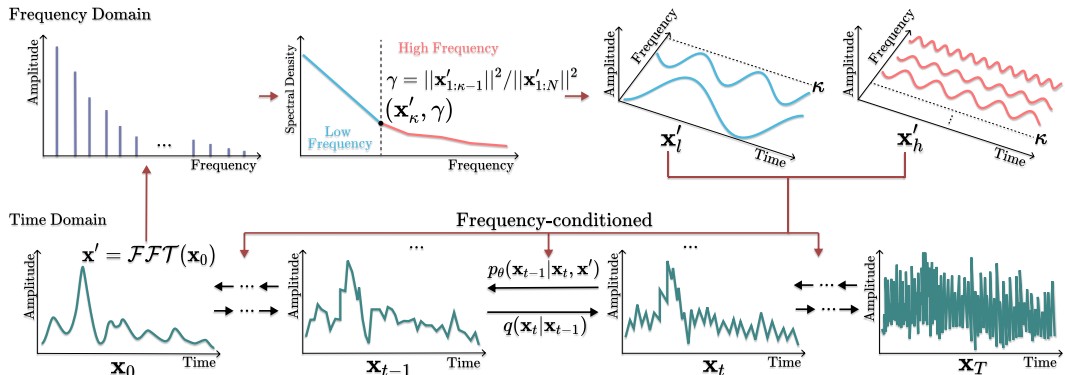

Figure 2: Illustration of the overall framework. Initially, the input value $\mathbf{x}_0$ is transformed into the frequency domain through a Fourier transformation, followed by the calculation of spectral density. Utilizing this information, our method adaptively separates low and high frequencies, which are then employed as conditions during the denoising process. This approach facilitates the gradual synthesis of data while taking into account both trends and semantic information.

information. In contrast, our proposed method effectively utilizes both global and semantic information from the frequency domain. Our proposed method is composed of two main components: i) frequency dissection for separate low-frequency components, which contain global trends, from high-frequency components, which capture semantic details, and ii) integration via cross-attention into a transformer-based decoder. We illustrate the overall procedures in Figure 2.

## 4.1 SIGNAL ENERGY-BASED ADAPTIVE FREQUENCY SELECTION

The distinction between low and high frequencies in time series analysis lacks a uniform standard, and prior methods often rely on arbitrary decisions or domain-specific knowledge (Lee et al., 2023). In our method, we address this limitation by calculating the spectral density based on signal energy, using it as an objective basis to separate low and high frequencies. This approach allows for a more data-driven separation within the time series.

In this process, the spectral density at $\mathbf{x}'$ is calculated to determine the information context for each frequency. We then compute the ratio of each frequency's information content to the total spectral information. Based on this spectral information, the frequencies containing information up to the threshold $\gamma$, which is a hyperparameter, are divided into low frequency components $\mathbf{x}'_l$, and the remaining are designated as high frequency components $\mathbf{x}'_h$:

$$\mathbf{x}'_l = \text{Padding}(\mathbf{x}'_{1:\kappa}), \quad \mathbf{x}'_h = \text{Padding}(\mathbf{x}'_{\kappa+1:N}), \tag{4}$$

where $\kappa$ denotes the frequency component index that satisfies $||\mathbf{x}'_{1:\kappa-1}||^2/||\mathbf{x}'_{1:N}||^2 < \gamma \leq ||\mathbf{x}'_{1:\kappa}||^2/||\mathbf{x}'_{1:N}||^2$ and $\text{Padding}(\cdot)$ keeps the dimension of each samples. Then, the signal energy of low-frequency and high-frequency features are each encoded through an embedding layer. This allows the model to capture the essential characteristics of the frequency components before they are integrated into the subsequent stages of the model.

As is widely known, most data exhibit high density at low frequencies (Crabbé et al., 2024). Building on this, we analyzed how frequencies are separated under different $\gamma$ and found that in most datasets, the majority of the information is concentrated at the first frequency, with the remaining frequencies being more evenly distributed with low density. Based on this observation, we set the threshold $\gamma$ to find the range where the information decays rapidly for each dataset. The experiments can be found in the Appendix B.1.

## 4.2 TEMPORAL-SPECTRAL TRANSFORMER-BASED DECODER

In the denoising process, our model employs a transformer-based architecture as the backbone network. This enables the capture of both the correlations between channels and the temporal dynamics

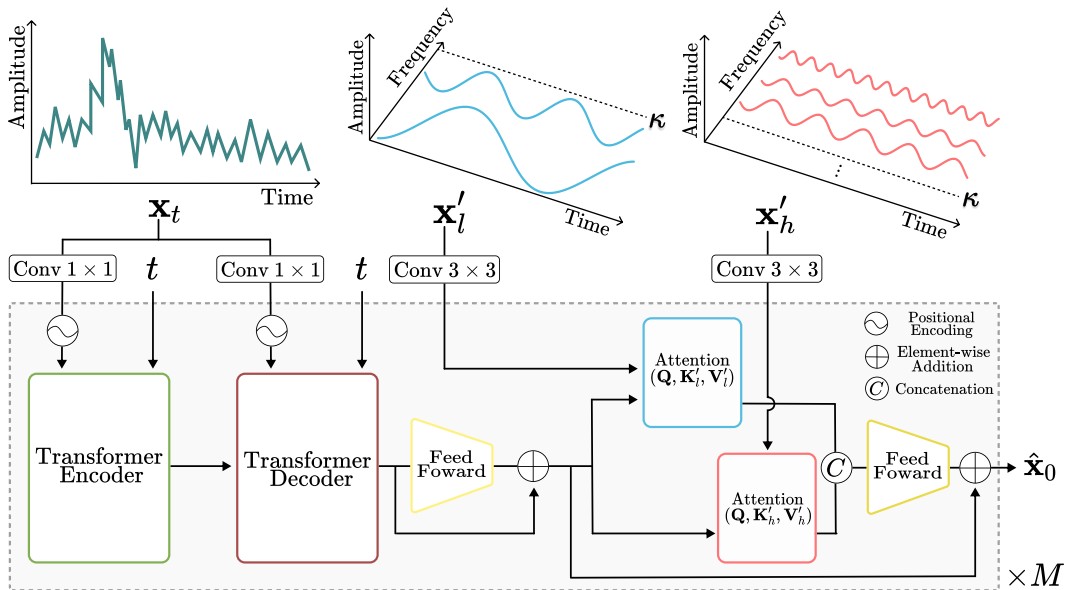

Figure 3: The architecture of the temporal-spectral transformer-based backbone network.

of time series data, effectively learning the inherent information in the time domain. By incorporating frequency information into this process, the model integrates valuable insights, enhancing performance not only in general generation tasks but also in more challenging scenarios such as long-range generation, imputation, and forecasting.

At diffusion step $t$, the input $\mathbf{x}_t$ first passes through a positional encoding to add temporal context, followed by a transformer block that captures temporal dynamics and correlations within the time series data: $\mathbf{z} = \text{TF}(\mathbf{x} + \text{PE})$ where $\text{PE}_{pos,2i} = \sin(\frac{pos}{10000^{2i/d}})$, $\text{PE}_{pos,2i+1} = \cos(\frac{pos}{10000^{2i/d}})$, and TF denotes transformer block. Simultaneously, the signal energy of frequency input $\mathbf{x}'_l$ and $\mathbf{x}'_h$ are processed through a 1D convolution encoder to extract features from the frequency domain, isolating global trends in $\mathbf{x}'_l$ and semantic details in $\mathbf{x}'_h$. The features processed in this manner are combined through cross-attention:

$$\mathbf{z}_l = \text{Attention}(\mathbf{Q}, \mathbf{K}'_l, \mathbf{V}'_l), \quad \mathbf{z}_h = \text{Attention}(\mathbf{Q}, \mathbf{K}'_h, \mathbf{V}'_h) \tag{5}$$

which enhances the denoising process and, as a result, improves the quality of the generated data. We depict decoder architecture in Figure 3.

### 4.3 TRAINING OBJECTIVE

**Frequency-conditioned DDPM loss.** To generate time series data based on frequency information, our network has to learn a frequency-conditioned objective loss function. Specifically, by extended the Equation 2, the following distribution is considered:

$$p_\theta(\mathbf{x}_{0:T}|\mathbf{x}') = p_\theta(\mathbf{x}_T) \prod_{t=1}^{T} q(\mathbf{x}_{t-1}|\mathbf{x}_t, \mathbf{x}'), \tag{6}$$

$$p_\theta(\mathbf{x}_{t-1}|\mathbf{x}_t, \mathbf{x}') = \mathcal{N}(\mathbf{x}_{t-1}; \mu_\theta(\mathbf{x}_t, \mathbf{x}', t), \Sigma_\theta(\mathbf{x}_t, \mathbf{x}', t)) \tag{7}$$

where $\mathbf{x}' = [\mathbf{x}'_l, \mathbf{x}'_h]$ and $\mathbf{x}_T \sim \mathcal{N}(0, \mathbf{I})$. Then, the reverse process is approximated by the following equation:

$$\mathbf{x}_{t-1} = \frac{\sqrt{\alpha_t}(1 - \bar{\alpha}_{t-1})}{1 - \bar{\alpha}_t} \mathbf{x}_t + \frac{\sqrt{\bar{\alpha}_{t-1}}\beta_t}{1 - \bar{\alpha}_t} \hat{\mathbf{x}}_0(\mathbf{x}_t, \mathbf{x}', t, \theta) + \frac{1 - \bar{\alpha}_{t-1}}{1 - \bar{\alpha}_t} \beta_t \epsilon_t, \tag{8}$$

where $\epsilon_t \sim \mathcal{N}(0, \mathbf{I})$, $\alpha_t = 1 - \beta_t$, and $\bar{\alpha}_t = \prod_{s=1}^{t} \alpha_s$. $\hat{\mathbf{x}}_0(\mathbf{x}_t, \mathbf{x}', t, \theta)$ denotes the predictive value for the original data generated by the conditional model. Finally, the network is trained through the

following objective function:

$$\mathcal{L}_{simple} = \mathbb{E}_{t,\mathbf{x}_0}\left[||\mathbf{x}_0 - \hat{\mathbf{x}}_0(\mathbf{x}_t, \mathbf{x}', t, \theta)||^2\right] \tag{9}$$

**Fourier-based loss.** Recent research on representation learning for time series has demonstrated that combining the time domain with the frequency domain yields superior results (Fons et al., 2022; Yuan & Qiao, 2024). In particular, our approach exploits frequency information as prior knowledge to capture the unique characteristics of the dataset itself, enhancing the model's representational power. This distinguishes it from previous methods that merely aim to improve frequency representation in sampled data, as our method directly utilizes frequency information. In line with these findings, our proposed method incorporates frequency information as a condition and employs a Fourier-based auxiliary loss to further enhance the performance of representation learning:

$$\mathcal{L}_{\mathcal{FFT}} = ||\mathcal{FFT}(\mathbf{x}_0) - \mathcal{FFT}(\hat{\mathbf{x}}_0(\mathbf{x}_t, \mathbf{x}', t, \theta))||^2. \tag{10}$$

Therefore, we define the overall objective function $\mathcal{L}_\theta$ as follows:

$$\mathcal{L}_\theta = \lambda_1 \mathcal{L}_{simple} + \lambda_2 \mathcal{L}_{\mathcal{FFT}} \tag{11}$$

where $\lambda_1$ and $\lambda_2$ are the hyperparameters to weight the corresponding losses.

## 5 EXPERIMENTS

In this section, we compared 6 competing models using 6 datasets to evaluate our proposed method. The comparison includes FourierDiffusion (Crabbé et al., 2024), Diffusion-TS (Yuan & Qiao, 2024), TimeGAN (Yoon et al., 2019), TimeVAE (Desai et al., 2021), Diffwave (Kong et al., 2021), and DiffTime (Coletta et al., 2024). To ensure a fair comparison, we replicated the experimental settings of Diffusion-TS (Yuan & Qiao, 2024) and TimeGAN (Yoon et al., 2019), taking the results for these models from (Yuan & Qiao, 2024). For a more detailed explanation of the experimental details, please refer to the Appendix A. The code is available at Github[1].

### 5.1 DATASETS

We utilized four real-world public datasets and two simulated datasets. **Stocks**: Daily stock data from Google (2004-2019) with six features including trading volume and various price metrics. **Energy**: A dataset from the UCI Appliances Energy prediction repository with 28 features related to household energy consumption, like temperature and humidity. **ETTh**$_1$: Records electricity transformer temperature data collected hourly, featuring oil temperature and six power load-related metrics. **fMRI**: A simulated BOLD time series dataset with 50 features, representing interactions between brain regions. **Sines**: A simulated multivariate dataset with five features generated from different frequencies and phases. **MuJoCo**: A dataset from the MuJoCo physics simulator with 14 features.

### 5.2 EVALUATION METRICS

In time series data generation, evaluation metrics focus on three key aspects: diversity (how well the model learns the data distribution), fidelity (how it captures temporal and spatial dependencies), and usefulness (its performance in prediction tasks). Accordingly, we employ the following 4 evaluation metrics:

**Discriminative score** (Yoon et al., 2019): Measures the similarity between real and generated data by evaluating whether they can be distinguished through supervised learning.

**Predictive score** (Yoon et al., 2019): Assess whether the generated data has captured predictive patterns by calculating the Mean Absolute Error (MAE) when using generated data to predict real data in the next time step.

**Context-Fréchet Inception Distance score (Context-FID score)** (Jeha et al., 2022): Determines how well local context has been captured by comparing the feature representations of real and generated data.

---
[1]https://anonymous.4open.science/r/Freq-Diff-E99F

**Correlational score** (Liao et al., 2020): Evaluates the temporal dependency similarity by calculating cross-correlations between real and generated time series data.

Table 1: Performance of time series generation task.

| Metric | Methods | Sines | Stocks | ETTh | MuJoCo | Energy | fMRI |
|---|---|---|---|---|---|---|---|
| Context-FID Score ↓ | Ours | **0.001**±**.000** | **0.024**±**.000** | **0.014**±**.000** | **0.004**±**.003** | **0.007**±**.000** | **0.071**±**.076** |
| | FourierDiffusion | 0.004±.000 | 0.052±.004 | 0.019±.001 | 0.083±.010 | 0.493±.031 | 0.121±.009 |
| | Diffusion-TS | 0.006±.000 | 0.147±.025 | 0.116±.010 | 0.013±.001 | 0.089±.024 | 0.105±.006 |
| | TimeGAN | 0.101±.014 | 0.103±.013 | 0.300±.013 | 0.563±.052 | 0.767±.103 | 1.292±.218 |
| | TimeVAE | 0.307±.060 | 0.215±.035 | 0.805±.186 | 0.251±.015 | 1.631±.142 | 14.449±.969 |
| | DiffTime | 0.006±.001 | 0.236±.074 | 0.299±.044 | 0.188±.028 | 0.279±.045 | 0.034±.015 |
| Correlational Score ↓ | Ours | **0.011**±**.002** | 0.007±.004 | **0.025**±**.007** | **0.192**±**.014** | **0.524**±**.028** | **0.628**±**.695** |
| | FourierDiffusion | 0.016±.002 | 0.015±.002 | 0.029±.005 | 0.243±.028 | 1.492±.141 | 1.216±.019 |
| | Diffusion-TS | 0.015±.004 | **0.004**±**.001** | 0.049±.008 | 0.193±.027 | 0.856±.147 | 1.411±.042 |
| | TimeGAN | 0.045±.010 | 0.063±.005 | 0.210±.006 | 0.886±.039 | 4.010±.104 | 23.502±.039 |
| | TimeVAE | 0.131±.010 | 0.095±.008 | 0.111±.020 | 0.388±.041 | 1.688±.226 | 17.296±.526 |
| | DiffTime | 0.017±.004 | 0.006±.002 | 0.067±.005 | 0.218±.031 | 1.158±.095 | 1.501±.048 |
| Discriminative Score ↓ | Ours | **0.005**±**.004** | **0.017**±**.016** | **0.006**±**.003** | **0.004**±**.003** | **0.012**±**.005** | **0.083**±**.077** |
| | FourierDiffusion | 0.009±.005 | 0.059±.064 | 0.010±.006 | 0.060±.007 | 0.241±.006 | 0.242±.013 |
| | Diffusion-TS | 0.006±.007 | 0.067±.015 | 0.061±.009 | 0.008±.002 | 0.122±.003 | 0.167±.023 |
| | TimeGAN | 0.011±.008 | 0.102±.021 | 0.114±.055 | 0.238±.068 | 0.236±.012 | 0.484±.042 |
| | TimeVAE | 0.041±.044 | 0.145±.120 | 0.209±.058 | 0.230±.102 | 0.499±.000 | 0.476±.044 |
| | DiffTime | 0.013±.006 | 0.097±.016 | 0.325±.099 | 0.426±.022 | 0.498±.002 | 0.492±.018 |
| Predictive Score ↓ | Ours | 0.094±.000 | **0.036**±**.000** | **0.119**±**.001** | 0.008±.001 | **0.249**±**.000** | **0.066**±**.032** |
| | FourierDiffusion | 0.094±.000 | **0.036**±**.000** | 0.120±.004 | 0.009±.000 | 0.251±.000 | 0.099±.000 |
| | Diffusion-TS | **0.093**±**.000** | **0.036**±**.000** | 0.119±.002 | **0.007**±**.000** | 0.250±.000 | 0.099±.000 |
| | TimeGAN | **0.093**±**.000** | 0.038±.001 | 0.124±.001 | 0.025±.003 | 0.273±.004 | 0.126±.002 |
| | TimeVAE | 0.093±.019 | 0.039±.000 | 0.126±.004 | 0.012±.002 | 0.292±.000 | 0.113±.003 |
| | DiffTime | **0.093**±**.000** | 0.038±.001 | 0.121±.004 | 0.010±.001 | 0.252±.000 | 0.100±.000 |
| | Original | 0.094±.001 | 0.036±.001 | 0.121±.005 | 0.007±.001 | 0.250±.003 | 0.090±.001 |

## 5.3 EXPERIMENTAL RESULTS AND ANALYSIS

### 5.3.1 TIME SERIES GENERATION

We conducted an experiment to generate time series data with a length of 24, a setting commonly used in baselines. As shown in Table 1, the proposed method outperforms other models in most datasets. Notably, the context-FID score shows excellent performance compared to baselines such as Diffusion-TS, indicating that the frequency information extracted through our proposed module is effectively leveraged in the diffusion learning process. Additionally, the discriminative score also demonstrated strong results, suggesting that high-frequency components played a significant role. Nevertheless, the predictive score showed similar performance to the comparative models, due to the relatively short time length of 24 used in this experiment.

Table 2: Performance of long-term time series data generation with 64, 128, and 256 lengths.

| Dataset | | Length | Ours | Diffusion-TS | TimeGAN | TimeVAE | Diffwave | DiffTime |
|---|---|---|---|---|---|---|---|---|
| ETTh | Discriminative Score ↓ | 64 | **0.010**±**.007** | 0.106±.048 | 0.227±.078 | 0.171±.142 | 0.254±.074 | 0.150±.003 |
| | | 128 | **0.009**±**.003** | 0.144±.060 | 0.188±.074 | 0.154±.087 | 0.274±.047 | 0.176±.015 |
| | | 256 | **0.021**±**.017** | 0.060±.030 | 0.442±.056 | 0.178±.076 | 0.304±.068 | 0.243±.005 |
| | Predictive Score ↓ | 64 | **0.081**±**.003** | 0.116±.000 | 0.132±.008 | 0.118±.004 | 0.133±.008 | 0.118±.004 |
| | | 128 | **0.074**±**.005** | 0.110±.003 | 0.153±.014 | 0.113±.005 | 0.129±.003 | 0.120±.008 |
| | | 256 | **0.071**±**.006** | 0.109±.013 | 0.220±.008 | 0.110±.027 | 0.132±.001 | 0.118±.003 |
| Energy | Discriminative Score ↓ | 64 | **0.068**±**.014** | 0.078±.021 | 0.498±.001 | 0.499±.000 | 0.497±.004 | 0.328±.031 |
| | | 128 | **0.128**±**.028** | 0.143±.075 | 0.499±.001 | 0.499±.000 | 0.499±.001 | 0.396±.024 |
| | | 256 | **0.257**±**.021** | 0.290±.123 | 0.499±.000 | 0.499±.000 | 0.499±.000 | 0.437±.095 |
| | Predictive Score ↓ | 64 | **0.242**±**.000** | 0.249±.000 | 0.291±.003 | 0.302±.001 | 0.252±.001 | 0.252±.000 |
| | | 128 | **0.241**±**.001** | 0.247±.001 | 0.303±.002 | 0.318±.000 | 0.252±.000 | 0.251.±.000 |
| | | 256 | **0.238**±**.002** | 0.245±.001 | 0.351±.004 | 0.353±.003 | 0.251±.000 | 0.251±.000 |

To further evaluate the effectiveness of low-frequency components in long-range generation tasks, we conducted additional experiments on time series data with lengths of 64, 128, and 256 using

ETTh and Energy datasets, as shown in Table 2. The proposed method demonstrated excellent performance across most metrics, with the predictive score being particularly noteworthy. In baseline models, as the time length increases to 64, 128, and 256, the challenge of capturing all information in the time domain makes learning more difficult. However, in our proposed method, the performance actually improved with longer time lengths. This improvement is primarily attributed to the incorporation of frequency information, especially the low-frequency components that capture global trends, enhancing the model's ability to generate long-range data effectively.

Additionally, we employed t-SNE (Van der Maaten & Hinton, 2008), principal component analysis (PCA) (Bryant & Yarnold, 1995), and kernel density estimation to visualize the performance of the generated data and assess whether the model distribution was well trained. As shown in Figure 4, our proposed method demonstrates a closer alignment with the distribution of the original dataset compared to TimeGAN in stock data, showing more similar visual patterns. This indicates that our model effectively learns the underlying original data distribution. More visualization samples can be found in the Appendix C.

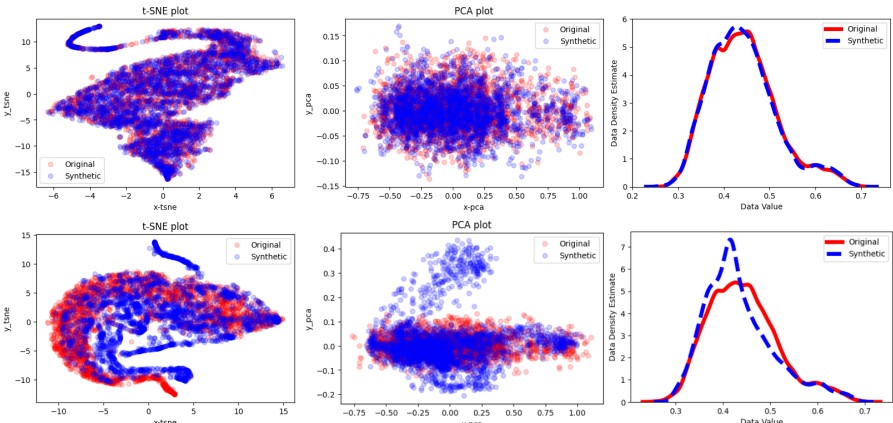

Figure 4: Visualization of synthetic time series compared to ours (top) and TimeGAN (bottom) on the energy dataset.

### 5.3.2 TASK-SPECIFIC GENERATION

The proposed frequency-based prior knowledge enhances the model's ability to capture both global and semantic information, which is crucial for improving accuracy and data quality, particularly in time series tasks such as imputation and forecasting. To evaluate this, we conducted additional experiments as illustrated in Figure 5, following (Yuan & Qiao, 2024) settings. For imputation, we assessed the model's performance across missing ratios of 10%, 25%, 50%, 75%, and 90%. Similarly, for forecasting, we evaluated the model with predicting sequence lengths of 6, 12, 24, and 36, over a total sequence length of 48 time steps. In the visualizations, the red cross indicates the observed values, the blue circle represents the ground truth, while the predicted values of the proposed method and the baseline are shown in green and gray, respectively, with the confidence interval represented by the shaded area. When analyzing both the visualization and quantitative results for imputation (top) and forecasting (bottom), it is evident that the proposed method achieves superior performance. Moreover, the performance gap widens as the missing ratio increases and the forecasting window extends, demonstrating the robustness of the proposed approach under more challenging conditions.

### 5.3.3 ABLATION STUDY

We confirmed through prior experiments that incorporating frequency information significantly enhances the synthesis of time series data. Specifically, as shown in Table 3, we evaluated different scenarios: using only low-frequency or high-frequency information, combining frequencies without division, and disregarding frequency information altogether. In the "w/o low frequency" case, where only high-frequency (semantic) information was considered, the discriminative score, particularly

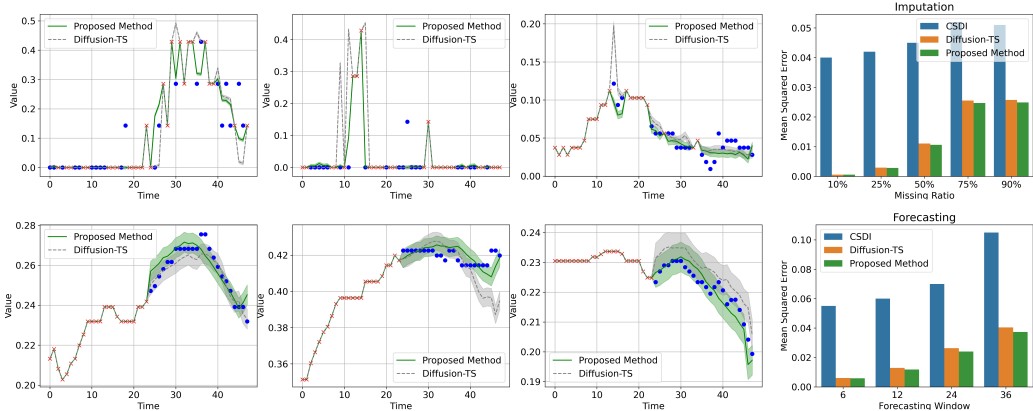

Figure 5: Visualization and empirical results of imputation (top) and prediction (bottom) on the energy dataset.

in the stocks and ETTh datasets, outperformed the "w/o high frequency" scenario. Moreover, when frequencies were modeled separately, the overall performance was superior compared to when both frequency components were modeled together (w/o adaptive). Importantly, incorporating both low and high-frequency information consistently resulted in better performance compared to models that did not utilize frequency information at all (w/o frequency).

Table 3: Ablation study results for frequency information.

| Metric | Methods | Sines | Stocks | ETTh | MuJoCo | Energy | fMRI |
|---|---|---|---|---|---|---|---|
| Discriminative Score ↓ | Ours | **0.005**±.004 | 0.017±.016 | **0.006**±.003 | **0.004**±.003 | **0.012**±.005 | **0.083**±.077 |
| | w/o low frequency | 0.007±.002 | 0.017±.011 | 0.009±.006 | 0.009±.008 | 0.058±.020 | 0.272±.149 |
| | w/o high frequency | **0.005**±.004 | 0.024±.014 | 0.007±.003 | 0.016±.008 | 0.014±.005 | 0.282±.065 |
| | w/o adaptive | **0.005**±.004 | **0.012**±.008 | 0.009±.006 | 0.007±.004 | 0.014±.008 | 0.221±.092 |
| | w/o frequency | 0.015±.010 | 0.115±.013 | 0.077±.005 | 0.023±.005 | 0.180±.016 | 0.259±.070 |
| Predictive Score ↓ | Ours | **0.094**±.000 | **0.036**±.000 | **0.119**±.001 | **0.008**±.000 | **0.249**±.000 | **0.066**±.032 |
| | w/o low frequency | **0.094**±.000 | 0.037±.000 | 0.120±.002 | 0.008±.001 | 0.250±.000 | 0.102±.000 |
| | w/o high frequency | **0.094**±.000 | **0.036**±.000 | 0.119±.005 | **0.008**±.000 | **0.249**±.000 | 0.101±.000 |
| | w/o adaptive | **0.094**±.000 | 0.037±.000 | 0.120±.002 | 0.008±.001 | **0.249**±.000 | 0.102±.000 |
| | w/o frequency | 0.095±.000 | 0.038±.000 | 0.123±.001 | **0.008**±.000 | 0.251±.000 | 0.101±.000 |
| | Original | 0.094±.001 | 0.036±.001 | 0.121±.005 | 0.007±.001 | 0.250±.003 | 0.090±.001 |

## 6 CONCLUSION

In this study, we present a novel diffusion model specifically designed for time series generation, leveraging frequency information as a priori knowledge. Our approach involves dissecting the frequency components into low and high frequencies based on spectral density, which allows the model to effectively capture global trends and local semantic details inherent in the data. The proposed method operates by integrating time-domain information during the denoising process, enabling it to generate data that reflects diverse perspectives from both the frequency and time domains. To validate our approach, we conducted extensive experiments on various public datasets, demonstrating the effectiveness and robustness of our method. We also performed an ablation study to assess the contributions of each frequency component, revealing how low and high frequencies impact the model's performance. Our results indicate that the proposed model excels not only in generating high-quality time series data but also in task-specific applications, such as imputation and forecasting, surpassing existing methods in versatility and accuracy. Our future direction is to expand toward the development of a foundation model. Through previous experiments, we have validated the benefits of incorporating frequency information. Building on this, our goal is to create a model that controls data generation using frequency-based conditions, learning from a variety of datasets, and performing tasks tailored to each dataset's specific characteristics.

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
