# A EXPERIMENTS DETAILS

## A.1 DATASETS

We used 4 real-world public datasets and 2 simulated datasets. **Stocks** is a daily stock data from Google spanning from 2004 to 2019 with 6 features, such as trading volume, high, low, opening, closing, and adjusted closing prices **Energy** is a dataset from the UCI Appliances Energy prediction repository. It contains 28 features related to household energy consumption, such as temperature, humidity, and energy usage, among others. **ETTh$_1$** is a dataset that records electricity transformer temperature data, which serves as a crucial indicator for long-term power system deployment. This dataset is collected at 1-hour intervals, with each data point consisting of the oil temperature and six power load-related features. **fMRI** is a simulated blood-oxygen-level-dependent (BOLD) time series dataset designed to estimate brain networks by analyzing interactions between nodes. In our experiments, we used data with 50 features, representing the activity of different brain regions or nodes over time. **Sines** is a simulated multivariate time series dataset consisting of five features. Each feature is generated using different frequencies and phases, allowing for a variety of periodic behaviors within the dataset. **MuJoCo** is a dataset generated using the MuJoCo physics simulator, consisting of 14 features.

Table 4: Characteristics of dataset.

| Dataset | Dimension | Samples |
|---------|-----------|---------|
| Stocks | 6 | 3773 |
| Energy | 28 | 19711 |
| ETTh$_1$ | 7 | 17420 |
| fMRI | 50 | 10000 |
| Sines | 5 | 10000 |
| MuJoCo | 14 | 10000 |

## A.2 EVALUATION METRICS

In time series data generation, evaluation metrics typically focus on three main aspects: diversity (how well the model has learned the data distribution), fidelity (whether the model captures the temporal dependencies and spatial relationships), and usefulness (how well the model performs in prediction tasks). These aspects provide a comprehensive evaluation of the model's ability to generate realistic and useful time series data, ensuring it not only mimics the data distribution but also captures essential temporal and spatial dependencies while remaining applicable to downstream tasks like prediction. Accordingly, we employ the following 4 evaluation metrics:

- Discriminative score (Yoon et al., 2019): To assess similarity, a 2-layer LSTM-based post-hoc time series data classification model is trained. The model is tasked with classifying original data as *"real"* and generated data as *"fake"*. After labeling the data accordingly, the RNN model is trained to distinguish between the two classes. The classification error on the test dataset is then measured, which serves as an indicator of the fidelity of the generated data, reflecting how well the synthetic data replicates the characteristics of the real data.

- Predictive score (Yoon et al., 2019): In addition, to evaluate the predictive capability, which is a key characteristic of time series data, a post-hoc sequence prediction model using a 2-layer LSTM is trained on the synthetic datasets. This model is trained to predict the value at the next time point. Its performance is then evaluated on the original dataset, and the prediction accuracy is measured using the Mean Absolute Error (MAE), providing insights into how well the synthetic data has captured the temporal dependencies of the original data.

- Context-FID score (Jeha et al., 2022): Context-FID is a modified version of the Fréchet Inception Distance (FID), originally used in image generation tasks but adapted for time series data where direct application is challenging. A lower Context-FID score indicates

a closer match between the distributions of real and generated data, which positively impacts the performance of downstream tasks. Specifically, instead of using Inception V3 (as in image-based FID), we leverage TS2Vec (Yue et al., 2022), a time series representation learning model, to derive embeddings for the data. The FID score is then calculated using these TS2Vec-encoded representations, enabling an effective similarity comparison for time series.

- Correlational score (Liao et al., 2020): To measure the change in correlation between variables over time in multivariate data, we compute a cross-correlation score. Specifically, the covariance between the $i-$th variable and the $j-$th variable is calculated as follows:

$$\text{cov}_{i,j} = \frac{1}{T} \sum_{t=1}^{T} \mathbf{X}_t^i \mathbf{X}_t^j - \Big(\frac{1}{T} \sum_{t=1}^{T} \mathbf{X}_t^i\Big)\Big(\frac{1}{T} \sum_{t=1}^{T} \mathbf{X}_t^j\Big), \tag{12}$$

where $T$ is the total time length. Then, the correlation is calculated:

$$\text{corr}_{i,j} := \frac{\text{cov}_{i,j}}{\sqrt{\text{cov}_{i,i}}\sqrt{\text{cov}_{j,j}}} \tag{13}$$

Finally, the MAE of the correlation between the real and generated data is computed. This cross-correlation score captures how the relationship between the two variables evolves over time.

## A.3 EXPERIMENTS SETUP

**Model details** As described in Section 4, our proposed method consists primarily of a signal energy-based frequency selection module and a backbone network. The backbone network, as depicted in Figure 3, follows an encoder-decoder structure augmented by temporal-spectral attention. The encoder is built using self-attention, feed-forward layers, and activation function layers, while the decoder incorporates self-attention and cross-attention layers applied to the encoded representations. The model performs cross-attention on the embedded frequencies, then concatenates these representations and passes them through a feed-forward layer to output the final $\hat{\mathbf{x}}_0$. During this process, the diffusion step $t$ is injected into the network, as seen in prior studies Ho et al. (2020); Yuan & Qiao (2024).

**Hyperparameter settings** We conducted a search for hyperparameter settings and selected the optimal configuration based on the discriminative score. The specific hyperparameters for model training, chosen for each dataset, are listed in Table 5. Additionally, the batch size was set to 128, and the learning rate was set to 1e-5. Regarding the hyperparameter gamma for frequency selection, as described in Section 4, most information is concentrated in low frequencies, so $\gamma$ was set to 0.8. With this setting, frequencies were separated in a ratio of 1:9, except for fMRI data, where the separation ratio was 3:7.

Table 5: Hyperparameter settings.

| Parameter | Sines | Stocks | ETTh | MuJoCo | Energy | fMRI |
|---|---|---|---|---|---|---|
| attention heads | 4 | 4 | 4 | 4 | 4 | 4 |
| attention dimension | 16 | 16 | 16 | 24 | 24 | 24 |
| encoder layers | 1 | 2 | 3 | 3 | 4 | 4 |
| decoder layers | 2 | 2 | 2 | 2 | 3 | 4 |
| timesteps | 500 | 500 | 500 | 1000 | 1000 | 1000 |
| training steps | 12000 | 12000 | 18000 | 20000 | 27000 | 20000 |
| $\gamma$ | 0.8 | 0.8 | 0.8 | 0.8 | 0.8 | 0.8 |

## B ADDITIONAL EXPERIMENTS

Our methodology can be broadly categorized into two key tasks: time-series data generation and task-specific generation, which includes imputation and forecasting. Both tasks follow the learning process illustrated in Figure 2, where the distribution of time-series data is modeled with consideration of frequency information. During training, complete data without missing values is input, sampled through a diffusion process, and output.

## B.1 TIME SERIES GENERATION

Due to space constraints, a simplified version of our experimental results is presented in Table 2 and Table 3 above. Hence, the full set of experimental results, which includes context-FID and correlational scores, can be found in Table 6 and Table 7.

Table 6: Performance of long-term time series data generation for all evaluation metrics.

| | Dataset | Length | Ours | Diffusion-TS | TimeGAN | TimeVAE | Diffwave | DiffTime |
|---|---|---|---|---|---|---|---|---|
| **ETTh** | Context-FID Score ↓ | 64 | **0.010±.001** | 0.631±.058 | 1.130±.102 | 0.827±.146 | 1.543±.143 | 1.279±.083 |
| | | 128 | **0.015±.001** | 0.787±.062 | 1.553±.169 | 1.062±.134 | 2.354±.170 | 2.554±.318 |
| | | 256 | **0.046±.004** | 0.423±.038 | 5.872±.208 | 0.826±.093 | 2.899±.289 | 3.524±.830 |
| | Correlational Score ↓ | 64 | **0.028±.009** | 0.082±.005 | 0.483±.019 | 0.067±.006 | 0.186±.008 | 0.094±.010 |
| | | 128 | **0.026±.012** | 0.088±.005 | 0.188±.006 | 0.054±.007 | 0.203±.006 | 0.222±.010 |
| | | 256 | **0.017±.006** | 0.064±.007 | 0.522±.013 | 0.046±.007 | 0.199±.003 | 0.135±.006 |
| | Discriminative Score ↓ | 64 | **0.010±.007** | 0.106±.048 | 0.227±.078 | 0.171±.142 | 0.254±.074 | 0.150±.003 |
| | | 128 | **0.009±.003** | 0.144±.060 | 0.188±.074 | 0.154±.087 | 0.274±.047 | 0.176±.015 |
| | | 256 | **0.021±.017** | 0.060±.030 | 0.442±.056 | 0.178±.076 | 0.304±.068 | 0.243±.005 |
| | Predictive Score ↓ | 64 | **0.081±.003** | 0.116±.000 | 0.132±.008 | 0.118±.004 | 0.133±.008 | 0.118±.004 |
| | | 128 | **0.074±.005** | 0.110±.003 | 0.153±.014 | 0.113±.005 | 0.129±.003 | 0.120±.008 |
| | | 256 | **0.071±.006** | 0.109±.013 | 0.220±.008 | 0.110±.027 | 0.132±.001 | 0.118±.003 |
| **Energy** | Context-FID Score ↓ | 64 | **0.011±.001** | 0.135±.017 | 1.230±.070 | 2.662±.087 | 2.697±.418 | 0.762±.157 |
| | | 128 | **0.019±.001** | 0.087±.019 | 2.535±.372 | 3.125±.106 | 5.552±.528 | 1.344±.131 |
| | | 256 | **0.010±.001** | 0.126±.024 | 5.032±.831 | 3.768±.998 | 5.572±.584 | 4.735±.729 |
| | Correlational Score ↓ | 64 | **0.493±.057** | 0.672±.035 | 3.668±.106 | 1.653±.208 | 6.847±.083 | 1.281±.218 |
| | | 128 | 0.556±.085 | **0.451±.079** | 4.790±.116 | 1.820±.329 | 6.663±.112 | 1.376±.201 |
| | | 256 | 0.504±.087 | **0.361±.092** | 4.487±.214 | 1.279±.114 | 5.690±.102 | 1.800±.138 |
| | Discriminative Score ↓ | 64 | **0.068±.014** | 0.078±.021 | 0.498±.001 | 0.499±.000 | 0.497±.004 | 0.328±.031 |
| | | 128 | **0.128±.028** | 0.143±.075 | 0.499±.001 | 0.499±.000 | 0.499±.001 | 0.396±.024 |
| | | 256 | **0.257±.021** | 0.290±.123 | 0.499±.000 | 0.499±.000 | 0.499±.000 | 0.437±.095 |
| | Predictive Score ↓ | 64 | **0.242±.000** | 0.249±.000 | 0.291±.003 | 0.302±.001 | 0.252±.001 | 0.252±.000 |
| | | 128 | **0.241±.001** | 0.247±.001 | 0.303±.002 | 0.318±.000 | 0.252±.000 | 0.251.±.000 |
| | | 256 | **0.238±.002** | 0.245±.001 | 0.351±.004 | 0.353±.003 | 0.251±.000 | 0.251±.000 |

Through ablation studies, we demonstrated that incorporating frequency information significantly enhances performance in the generation task. Moreover, adaptively separating low and high-frequency components, rather than relying solely on specific frequencies, proved to be more effective. Specifically, low-frequency components contribute to capturing global trends, improving overall prediction capabilities, while high-frequency components provide semantic details, enhancing fine-grained generation. This adaptive separation aligns with the inductive bias of the diffusion process, facilitating better data synthesis. Experimental results in Table 7 supported this claim: the predictive score was lower when low-frequency information was excluded, and the discriminative score suffered without high-frequency information due to the absence of semantic priors. Although experiments were conducted with a 24-window length, which had minimal impact on outcomes, the consistent results across trials reinforce our conclusions.

We additionally performed an ablation study on the hyperparameter $\gamma$. Specifically, we conducted experiments varying $\gamma$ from 0.8 (the original setting) down to 0.1, focusing particularly on the fMRI dataset, which exhibits a more evenly distributed power spectrum compared to other datasets. The results, presented in Table 8, demonstrate that the best performance is achieved with $\gamma$ set to 0.8. This indicates that the model performs optimally when the power spectrum reflects an 8:2 ratio, that is, when 80% of the information is concentrated in low frequencies.

## B.2 TASK-SPECIFIC GENERATION

For a task-specific generation, the conditional distribution is approximately sampled using the pre-trained diffusion model and the gradient of the classifier, following the sampling method in Diffusion-TS (Yuan & Qiao, 2024). However, frequency information, one of central to our method, is not directly accessible during inference. This limitation is particularly pronounced in data with missing values, where acquisition distortions prevent obtaining normal frequency information. To address this, we leverage the training dataset used for pre-training to calculate spectral density and use this frequency information during inference. In task-specific generation, the process involves

Table 7: Ablation study results for all evaluation metrics.

| Metric | Methods | Sines | Stocks | ETTh | MuJoCo | Energy | fMRI |
|---|---|---|---|---|---|---|---|
| Context-FID Score ↓ | Ours | **0.001**±**.000** | **0.024**±**.000** | **0.014**±**.000** | **0.004**±**.003** | 0.007±.000 | **0.071**±**.076** |
| | w/o low frequency | **0.001**±**.000** | 0.026±.001 | 0.021±.000 | 0.017±.001 | 0.011±.002 | 0.216±.012 |
| | w/o high frequency | 0.002±.000 | 0.025±.002 | 0.025±.000 | 0.009±.001 | 0.007±.000 | 0.193±.009 |
| | w/o adaptive | **0.001**±**.000** | 0.024±.001 | 0.023±.000 | 0.006±.001 | **0.006**±**.000** | 0.209±.017 |
| | w/o frequency | 0.010±.002 | 0.148±.017 | 0.166±.014 | 0.017±.000 | 0.140±.019 | 0.291±.009 |
| Correlational Score ↓ | Ours | **0.011**±**.002** | **0.004**±**.001** | 0.025±.007 | 0.192±.014 | 0.524±.028 | **0.628**±**.695** |
| | w/o low frequency | 0.014±.005 | 0.009±.003 | 0.027±.012 | 0.209±.036 | **0.438**±**.019** | 1.496±.022 |
| | w/o high frequency | 0.014±.003 | 0.013±.002 | **0.025**±**.004** | **0.180**±**.011** | 0.467±.099 | 1.256±.023 |
| | w/o adaptive | 0.018±.002 | 0.010±.002 | 0.027±.012 | 0.192±.016 | 0.469±.081 | 1.389±.023 |
| | w/o frequency | 0.015±.003 | 0.004±.003 | 0.055±.007 | 0.197±.031 | 0.936±.085 | 1.626±.032 |
| Discriminative Score ↓ | Ours | **0.005**±**.004** | 0.017±.016 | **0.006**±**.003** | **0.004**±**.003** | **0.012**±**.005** | **0.083**±**.077** |
| | w/o low frequency | 0.007±.002 | 0.017±.011 | 0.009±.006 | 0.009±.008 | 0.058±.020 | 0.272±.149 |
| | w/o high frequency | **0.005**±**.004** | 0.024±.014 | 0.007±.003 | 0.016±.008 | 0.014±.005 | 0.282±.065 |
| | w/o adaptive | **0.005**±**.004** | **0.012**±**.008** | 0.009±.006 | 0.007±.004 | 0.014±.008 | 0.221±.092 |
| | w/o frequency | 0.015±.010 | 0.115±.013 | 0.077±.005 | 0.023±.005 | 0.180±.016 | 0.259±.070 |
| Predictive Score ↓ | Ours | **0.094**±**.000** | **0.036**±**.000** | 0.119±.001 | **0.008**±**.000** | **0.249**±**.000** | **0.066**±**.032** |
| | w/o low frequency | **0.094**±**.000** | 0.037±.000 | 0.120±.002 | 0.008±.001 | 0.250±.000 | 0.102±.000 |
| | w/o high frequency | **0.094**±**.000** | **0.036**±**.000** | 0.119±.005 | **0.008**±**.000** | **0.249**±**.000** | 0.101±.000 |
| | w/o adaptive | **0.094**±**.000** | 0.037±.000 | 0.120±.002 | 0.008±.001 | **0.249**±**.000** | 0.102±.000 |
| | w/o frequency | 0.095±.000 | 0.038±.000 | 0.123±.001 | **0.008**±**.000** | 0.251±.000 | 0.101±.000 |
| | Original | 0.094±.001 | 0.036±.001 | 0.121±.005 | 0.007±.001 | 0.250±.003 | 0.090±.001 |

Table 8: Ablation study results for various $\gamma$ on the fMRI dataset.

| $\gamma$ | Context-FID Score ↓ | Correlation Score ↓ | Discriminative Score ↓ | Predictive Score ↓ |
|---|---|---|---|---|
| 0.8 | **0.071**±**.076** | **0.628**±**.695** | **0.083**±**.077** | **0.066**±**.032** |
| 0.6 | 0.148±.004 | 1.316±.017 | 0.149±.021 | 0.102±.000 |
| 0.4 | 0.193±.008 | 1.303±.038 | 0.108±.048 | 0.102±.000 |
| 0.3 | 0.177±.008 | 1.288±.028 | 0.133±.057 | 0.101±.000 |
| 0.2 | 0.161±.011 | 1.330±.034 | 0.132±.061 | 0.102±.000 |
| 0.1 | 0.154±.014 | 1.312±.042 | 0.125±.048 | 0.102±.000 |

inputting data with missing values alongside the frequency information extracted during training into the approximate sampling method of the conditional distribution via the pre-trained diffusion model. The output is a complete dataset with the missing values imputed or future values predicted, ensuring continuity and coherence in the data.

To enhance the reliability of the proposed method, we conducted additional imputation and forecasting experiments following the settings outlined in SSSD (Alcaraz & Strodthoff, 2023). For imputation, we evaluated performance at 70%, 80%, and 90% missing values on the MuJoCo dataset, with the results measured using MSE presented in Table 9. All MSE values are in the order of 1e-3, and the proposed method achieved the best performance at 80% and 90% missing rates.

For forecasting, we utilized the Solar dataset from GluonTS (Alexandrov et al., 2020), maintaining the same experimental settings as SSSD (Alcaraz & Strodthoff, 2023) to ensure a fair comparison in Table 10. Using 168 observations, the model generated predictions for the next 24 time steps. The proposed method demonstrated superior performance, particularly when compared to transformer-based models such as iTransformer and PatchTST. These results further confirm the efficacy and robustness of the proposed method for task-specific generation tasks.

## C ADDITIONAL VISUALIZATION

We provide additional visualizations in Figures 6 to 10. In Figure 6, alongside Figure 4, we include the results of t-SNE, PCA, and kernel density estimation for all datasets. Furthermore, in Figures 7 through 10, we present visualizations showcasing performance across different imputation ratios and forecasting windows.

Table 9: Performance of time series imputation on the MuJoCo dataset.

| Model | 70% Missing | 80% Missing | 90% Missing |
|---|---|---|---|
| RNN GRU-D | 11.34 | 14.21 | 19.68 |
| ODE-RNN | 9.86 | 12.09 | 16.47 |
| NeuralCDE | 8.35 | 10.71 | 13.52 |
| Latent-ODE | 3 | 2.95 | 3.6 |
| NAOMI | 1.46 | 2.32 | 4.42 |
| NRTSI | 0.63 | 1.22 | 4.06 |
| CSDI | **0.24(3)** | 0.61(10) | 4.84(2) |
| SSSD | 0.59(8) | 1.00(5) | 1.90(3) |
| Diffusion-TS | 0.37(3) | 0.43(3) | 0.73(12) |
| Ours | 0.31(4.5) | **0.35(5)** | **0.45(1.6)** |

Table 10: Performance of time series forecasting on the Solar dataset.

| Model | MSE |
|---|---|
| GP-copula | $9.8e2 \pm 5.2e1$ |
| TransMAF | $9.3e2$ |
| TLAE | $6.8e2 \pm 7.5e1$ |
| CSDI | $9.0e2 \pm 6.1e1$ |
| SSSD | $5.03e2 \pm 1.06e1$ |
| Diffusion-TS | $3.75e2 \pm 3.6e1$ |
| PatchTST | $3.80e2$ |
| iTransformer | $3.73e2$ |
| Ours | $\mathbf{3.41e2 \pm 1.4e1}$ |

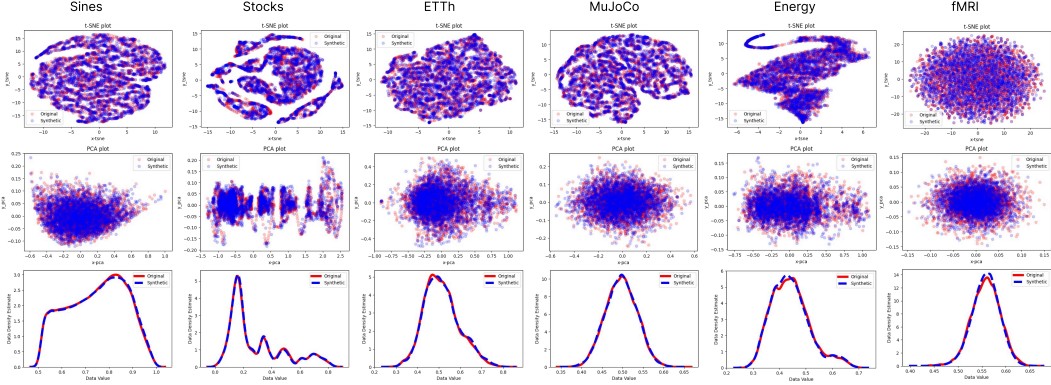

Figure 6: Visualization of synthetic time series.

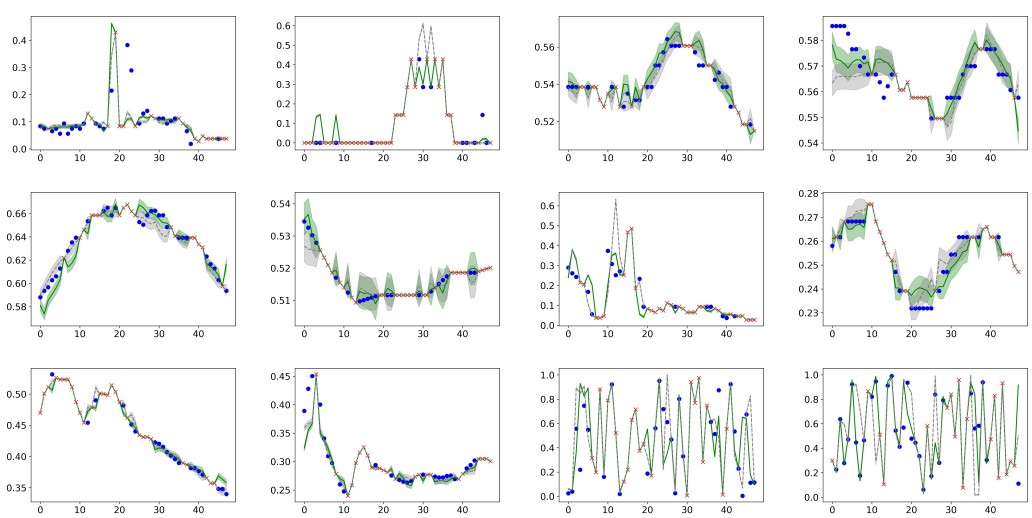

Figure 7: Visualization of imputation for 50% missing values on the energy dataset.

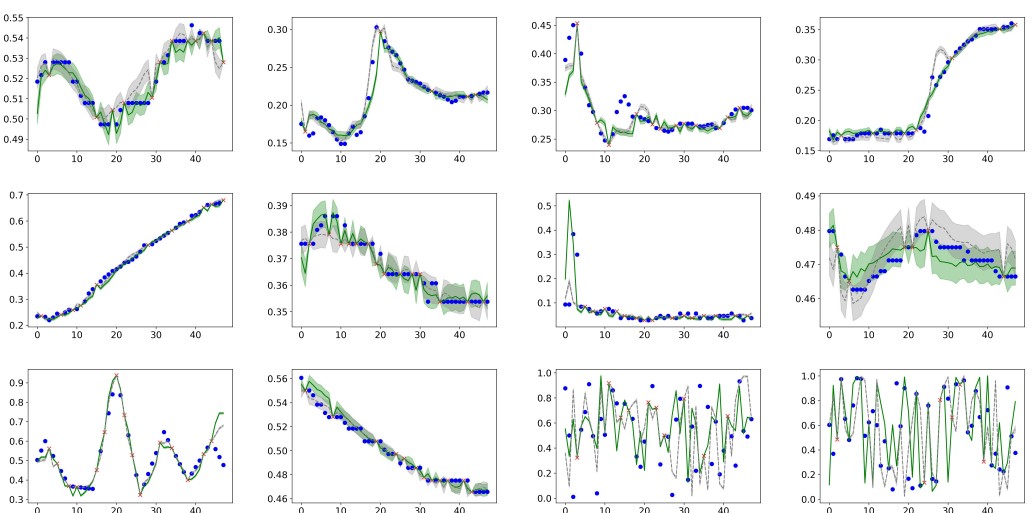

Figure 8: Visualization of imputation for 90% missing values on the energy dataset.

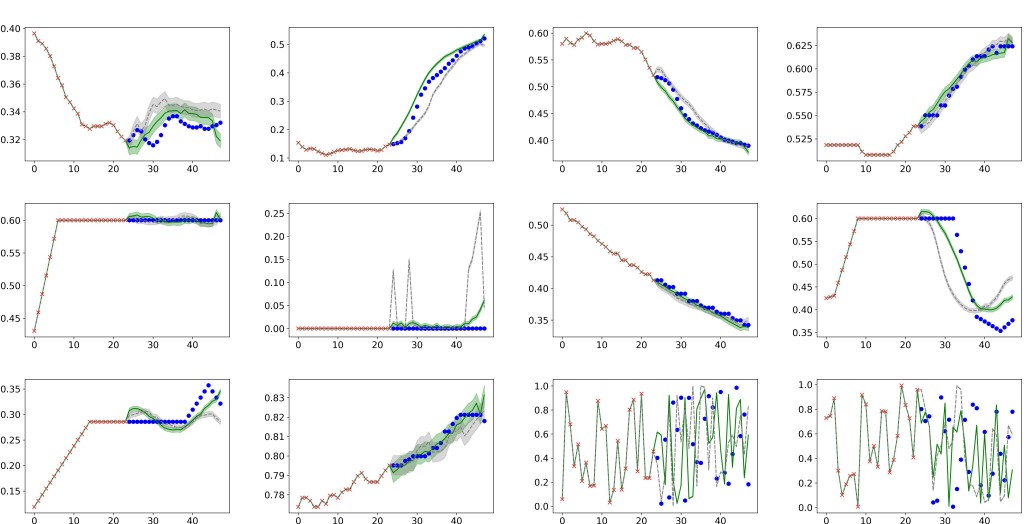

Figure 9: Visualization of forecasting results for sequence length of 24 on the energy dataset.

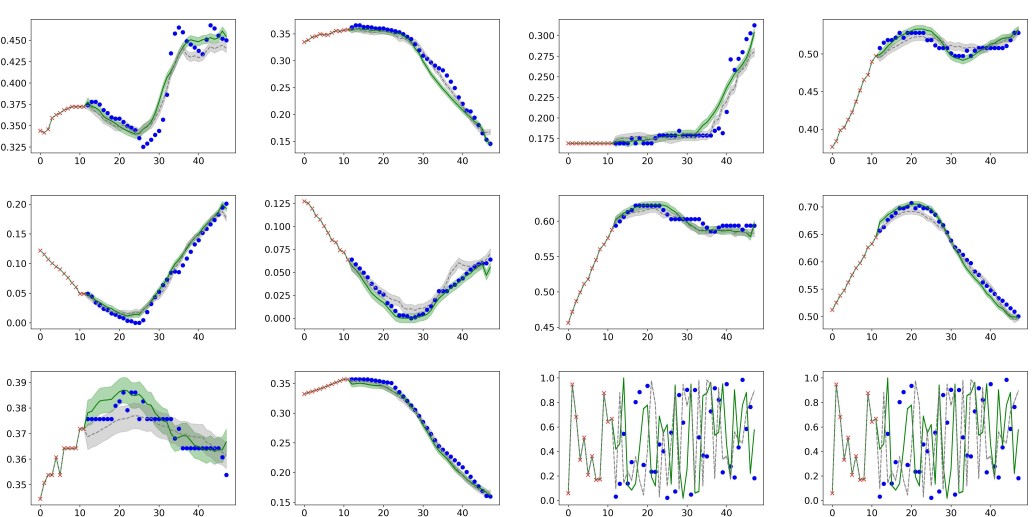

Figure 10: Visualization of forecasting results for sequence length of 36 on the energy dataset.