# OpenReview forum: "Frequency-Conditioned Diffusion Models for Time Series Generation"
_ICLR.cc/2025/Conference — Submitted to ICLR 2025_

### Official Review · Reviewer_mxUK · 2024-10-22

**Soundness:** 2
**Presentation:** 2
**Contribution:** 3
**Rating:** 5
**Confidence:** 3

**Summary:**

This paper introduces a new diffusion model that uses frequency domain information to improve time series data generation, and the proposed model outperforms some advanced methods in accuracy and flexibility.

**Strengths:**

1. The paper's introduction of frequency domain information into the time series diffusion model exhibits a certain degree of innovation.

2.The method proposed in the paper achieves superior performance on series generation tasks.

3.The paper is understandable in most parts.

**Weaknesses:**

1.Despite the existence of some related research, I remain uncertain about the practical applications of unconditional time series generation. I think that newly generated time series data is far less valuable than text or images produced by other generative tasks. Therefore, I place greater importance on conditional generation tasks such as forecasting and imputation. However, the experiments related to forecasting or imputation in the paper are not sufficiently comprehensive. The paper should evaluate and compare more advanced forecasting or imputation models, such as iTransformer and TimesNet, on a broader range of datasets.

2.The description of the method is not very clear. It is suggested that the authors use a pseudocode format to describe the training and inference processes of the proposed algorithm.

3.The paper states that “in the inference step, frequency information is directly extracted from the dataset and utilized”. Could this lead to information leakage?

The paper includes some statements that appear rather definitive without providing detailed explanations or citing sources. For instance: 1) those approaches make it nearly impossible to capture all time-related information, particularly in long-range sequences, making it a highly challenging problem；2) generative models struggle to effectively model the temporal-spatial-spectral information inherent in multivariate time series data, particularly during the denoising process, where capturing this complex information proves to be challenging.

**Questions:**

1.During the reverse process of the training phase, we restore the noisy series XT back to the original series X0. The frequency domain information of the noisy series XT is obviously of little value. Could the authors clarify how the proposed method specifically applies frequency domain information? Similarly, during the inference stage, extracting frequency domain information from the noisy series would be unhelpful, whereas extracting it from the original series would raise concerns about information leakage.

2.How does the forecasting performance of the model compare to the SOTA models, such as iTransformer and PatchTST?

3.How does the training and inference efficiency of the proposed method compare to other methods, such as Diffusion-TS?

---

> ### Author Response · Authors · 2024-11-24
> **Response to Reviewer mxUK**
>
> We appreciate the reviewer’s valuable comments. We address individual concerns below.
>
> **Response to Question 1)**
> In the denoising process, the objective is to reconstruct the noisy time series $x_T$ back to the original series $x_0$. During this process, frequency-domain information is incorporated as a conditional input to guide the reconstruction. As depicted in Figure 2, the frequency information used as a condition is derived from the original series $x_0$ and is employed during the training phase to enhance the denoising accuracy.
>
> During the inference step, frequency information is sampled from the training dataset and utilized. To achieve this, we employ kernel density estimation, a nonparametric approach, to estimate the distribution by analyzing the spectral density of the training data. This estimated distribution provides the necessary frequency information for inference, ensuring consistency with the characteristics of the training dataset.
>
> **Response to Question 2)**
> We appreciate the reviewer’s suggestion, as it provides an opportunity to further validate the robustness of our proposed method. In response, we conducted additional forecasting experiments using the Solar dataset [1], predicting the next 24 windows based on 168 time steps, in alignment with the experimental settings outlined in SSSD [2].
> | Model         | MSE                 |
> |---------------|---------------------|
> | GP-copula     | 9.8e2±5.2e1         |
> | TransMAF      | 9.3e2               |
> | TLAE          | 6.8e2±7.5e1         |
> | CSDI          | 9.0e2±6.1e1         |
> | SSSD          | 5.03e2±1.06e1       |
> | Diffusion-TS  | 3.75e2±3.6e1        |
> | PatchTST      | 3.80e2              |
> | iTransformer  | 3.73e2              |
> | Ours          | **3.41e2±1.4e1**    |
> The results show that our proposed method achieves the best performance compared to both diffusion-based methods and Transformer-based methods. This highlights the efficiency of our approach not only in data generation but also in task-specific generation scenarios such as forecasting. These additional experimental results will be included in Appendix B.2 of the revised paper.
>
> **Response to Question 3)**
> The table below shows the model parameters and flops of our model and Diffusion-TS on the stock dataset.
> | Model         | # Parameters | # FLOPs  |
> |---------------|--------------|----------|
> | Diffusion-TS  | 0.316M       | 6.58M    |
> | Ours          | 0.455M       | 9.025M   |
> The proposed method has a slightly higher number of parameters compared to Diffusion-TS, owing to the encoder used for frequency information extraction and the cross-attention mechanism. However, the difference in FLOPs is only 3M, and this has minimal impact on both learning and inference speed, indicating that the method achieves performance improvements without compromising efficiency.
>
> **Reference**
>
> [1] Alexandrov et al., “Gluonts: Probabilistic and neural time series modeling in python,” JMLR, 2020.
>
> [2] Alcaraz et al., “Diffusion-based time series imputation and forecasting with structured state space models,” TMLR, 2022

---

### Official Review · Reviewer_8Qz6 · 2024-11-01

**Soundness:** 1
**Presentation:** 2
**Contribution:** 1
**Rating:** 3
**Confidence:** 4

**Summary:**

This paper proposes a frequency-conditioned diffusion model to conduct time series generation tasks including forecasting and imputation. The author believes that previous methods did not fully utilize  frequency domain information, leading to poor modeling of low-frequency signals such as trends. As a result, the proposed method utilize FFT to extract frequency domain information and models low-frequency and high-frequency information separately during denoising. The authors claim that the proposed method achieves state-of-the-art performances under multiple evaluation settings.

**Strengths:**

1. The network design for joint utilization of time and frequency domain information is reasonable. The authors uses a cross attention structure for frequency information modeling, which is scalable and reasonable.

2. The energy-based adaptive frequency selection strategy is novel. The authors determine the division of low-frequency and high-frequency parts by spectral energy. This principle is transferable across different data sets and holds certain value for subsequent research based on frequency domain information.

**Weaknesses:**

1. The settings of most experiments are not well explained. For example, in the generation task, how is the noise sampling conducted? Is there a reasonable control to ensure sampling consistency across different methods?

2.The authors did not provide detailed descriptions of how the proposed method performs conditional generation such as forecasting and permutation. The formulation in Sec. 4 only describes unconditional generation.

**Questions:**

**Information leakage**: The frequency information used by denoising model is extracted from the original sample
$x_0$ rather than the noisy $x_t$ (as seen in line #241 of diffusion.py), which undoubtedly cause information leakage.  Consequently, The experimental results presented in  this article are meaningless.

---

> ### Author Response · Authors · 2024-11-23
> **Response to Reviewer 8Qz6**
>
> **Response to Question**
>
> Thank you for pointing this out. It seems there may have been a misunderstanding regarding the role of frequency information in our proposed method. The reviewer observed that frequency information was extracted from the original $x_0$ during the denoising process. However, our intention was to conceptualize the condition as a form of prior knowledge, consistent with prior work [1, 2]. By design, this prior serves as a meaningful guide in the denoising process.
>
> Extracting frequency information from the noisy $x_T$, given its inherently degraded nature, would not provide reliable guidance. Therefore, we deliberately chose to derive frequency information from $x_0$ to ensure that the condition introduced during denoising was robust and meaningful. This approach aligns with our goal of leveraging prior knowledge to improve the effectiveness of the denoising process.
>
> **Reference**
>
> [1] Galib et al., “FIDE: Frequency-inflated conditional diffusion model for extreme-aware time series generation,” NeurIPS, 2024
>
> [2] Narasimhan et al., “Time weaver; A conditional time series generation model,” ICML, 2024

---

### Official Review · Reviewer_cjt7 · 2024-11-02

**Soundness:** 2
**Presentation:** 3
**Contribution:** 2
**Rating:** 5
**Confidence:** 4

**Summary:**

This paper presents a diffusion model for generating time series data. The model improves upon existing methods by incorporating frequency domain information, specifically separating low-frequency global trends from high-frequency details. This separation allows the model to better capture important patterns during the denoising process. The model then uses a specialized frequency encoder to integrate this information, improving the model's ability to capture both global and local features. The paper evaluates the model's performance on several public datasets and compares it to existing models, demonstrating its effectiveness in generating time series data for diverse tasks such as forecasting and imputation. The paper is well-structured and easy to understand.

**Strengths:**

(1) The proposed generation diffusion model operates in the time domain while concurrently leveraging frequency domain information as a priori knowledge, allowing to capture and represent the data's characteristics by combining information from both domains during the diffusion learning process.
(2) The adaptive dissects of frequency information based on the power spectrum enables the model to represent the temporal aspects embedded within the data, leading to the synthesis of high-quality data that more accurately reflects the trends and patterns present within the dataset.
(3) Experiments across multiple datasets showcase the model's versatility across diverse time series generation challenges where it achieves superior performance in long-range data generation, imputation and forecasting tasks.

**Weaknesses:**

(1) While incorporating frequency domain information and a Transformer-based decoder can enhance model performance, it also introduces additional complexity and computational overhead compared to simpler approaches. A thorough comparative analysis with baseline models is necessary to assess the trade-off between performance gains and increased computational cost.
(2) A more detailed explanation of the frequency encoder's architecture would enhance the paper's clarity. While the use of a 1D convolutional encoder is noted, a deeper discussion on the reasons for selecting this specific architecture and its suitability for capturing frequency-domain information would be valuable.
(3) While the use of a threshold parameter (\gamma) to separate low and high-frequency components based on spectral density is a reasonable approach, a more detailed analysis is needed to assess the sensitivity of the model's performance to variations in this parameter.

**Questions:**

(1) The authors assert that exclusive reliance on frequency domain modeling is suboptimal due to the loss of temporal information. However, this claim is contradicted by Diffusion-TS, which effectively leverages Fourier basis functions to capture the crucial seasonal component of time series. Furthermore, Carabbe et al. [Time Series Diffusion in the Frequency Domain. (ICML 2024)] empirically demonstrate that frequency diffusion models exhibit superior performance in capturing the training distribution compared to their time-domain counterparts. This is attributed to the inherent localization of time series data in the frequency domain, rendering it more amenable to modeling in this space. To strengthen the core argument of this work, a more detailed elaboration on this point is essential.

(2) The majority of baseline models employed in the comparative analysis are time-domain based, limiting the scope of direct comparison with purely frequency-domain diffusion models. Additionally, the ablation study lacks a variant that exclusively operates in the frequency domain. This would help to show if combining both ways is really better than using each one separately.

(3) In the inference phase, how do the randomly initialized noise samples acquire the necessary frequency information to generate diverse and realistic samples? Moreover, how can we ensure that the generation process covers the entire spectrum of frequencies present in the target data distribution?

(4) In Section 4.1, the authors posit that their analysis reveals a concentration of information primarily in the first frequency component, with subsequent frequencies exhibiting a more even distribution of lower density. However, this claim lacks empirical validation. To substantiate this assertion, the authors should conduct experiments with varying values of γ to demonstrate the impact on model performance.

(5) How does the utilization of a Transformer decoder contribute to the model's performance? Could a simpler encoder-only architecture suffice to reduce computational complexity?

(6) In Table 1, what is the reference point for the "original" values? While the proposed model exhibits comparable performance to Diffusion-TS across most datasets, it demonstrates a significant improvement on the fMRI dataset. Is there any explanation to this?

---

> ### Author Response · Authors · 2024-11-23
> **Response to Reviewer cjt7 (Part 1)**
>
> We appreciate the reviewer’s valuable comments. We address individual concerns below.
>
> **Response to Question 1)**
> Diffusion-TS [1] employs Fourier basis functions to capture seasonal patterns but ultimately generates data by combining trend information from the time domain with representations processed through its decoder module. This demonstrates that Diffusion-TS relies on a hybrid approach, integrating both the time and frequency domains rather than depending solely on one. Similarly, while FourierDiffusion [2] advocates for the superiority of frequency-domain modeling, analysis in Section 4 reveals that the datasets used in their experiments exhibited stronger localization in the spectral domain compared to the temporal domain. This highlights that the importance of temporal versus spectral components varies based on dataset characteristics. To address this variability, we adopted a multi-view modeling approach that combines the strengths of both domains, aiming to achieve optimal performance regardless of the dataset’s dominant characteristics.
>
> **Response to Question 2)**
> We appreciate the reviewer’s comments, which could contribute to further enhancing the robustness of our proposed method. However, the proposed method fundamentally operates in the time domain while utilizing the frequency domain as conditional prior knowledge. As a result, performance comparisons solely within the frequency domain are inherently limited. Hence, we included only a performance comparison in the time domain (w/o frequency in Table 3) to provide a comprehensive evaluation.
>
> That said, we acknowledge the reviewer’s valuable perspective on the potential advantages of pure frequency-domain diffusion models in demonstrating the robustness of our approach. In response, we conducted comparative generation experiments using the FourierDiffusion [2] model as a baseline, with the results presented in the table below.
> | Metric             | Methods           | Sines           | Stocks          | ETTh            | MuJoCo          | Energy          | fMRI            |
> |--------------------|-------------------|-----------------|-----------------|-----------------|-----------------|-----------------|-----------------|
> | **Context-FID Score ↓** | Ours              | **0.001±.000**  | **0.024±.000**  | **0.014±.000**  | **0.004±.003**  | **0.007±.000**  | **0.071±.076**  |
> |                    | FourierDiffusion  | 0.004±.000      | 0.052±.004      | 0.019±.001      | 0.083±.010      | 0.493±.031      | 0.121±.009      |
> |                    | Diffusion-TS      | 0.006±.000      | 0.147±.025      | 0.116±.010      | 0.013±.001      | 0.089±.024      | 0.105±.006      |
> | **Correlational Score ↓** | Ours              | **0.011±.002**  | 0.007±.004      | **0.025±.007**  | **0.192±.014**  | **0.524±.028**  | **0.628±.695**  |
> |                    | FourierDiffusion  | 0.016±.002      | 0.015±.002      | 0.029±.005      | 0.243±.028      | 1.492±.141      | 1.216±.019      |
> |                    | Diffusion-TS      | 0.015±.004      | **0.004±.001**  | 0.049±.008      | 0.193±.027      | 0.856±.147      | 1.411±.042      |
> | **Discriminative Score ↓** | Ours              | **0.005±.004**  | **0.017±.016**  | **0.006±.003**  | **0.004±.003**  | **0.012±.005**  | **0.083±.077**  |
> |                    | FourierDiffusion  | 0.009±.005      | 0.059±.064      | 0.010±.006      | 0.060±.007      | 0.241±.006      | 0.242±.013      |
> |                    | Diffusion-TS      | 0.006±.007      | 0.067±.015      | 0.061±.009      | 0.008±.002      | 0.122±.003      | 0.167±.023      |
> | **Predictive Score ↓** | Ours              | 0.094±.000      | **0.036±.000**  | **0.119±.001**  | 0.008±.001      | **0.249±.000**  | **0.066±.032**  |
> |                    | FourierDiffusion  | 0.094±.000      | **0.036±.000**  | 0.120±.004      | 0.009±.000      | 0.251±.000      | 0.100±.000      |
> |                    | Diffusion-TS      | **0.093±.000**  | **0.036±.000**  | 0.119±.002      | **0.007±.000**  | 0.250±.000      | 0.099±.000      |
> |                    | Original          | 0.094±.001      | 0.036±.001      | 0.121±.005      | 0.007±.001      | 0.250±.003      | 0.090±.001      |
>
> The results show that FourierDiffusion generally performs well compared to time-domain diffusion models, which highlights the inherent importance of frequency-domain representations in time series data. However, FourierDiffusion exhibited comparatively lower performance on datasets like Energy, underscoring the limitations of relying solely on the frequency domain for modeling. This demonstrates the importance of incorporating time-domain information and validates the robustness of our proposed multi-view method, which effectively integrates both domains for optimal performance. The results of the experiments have been added to Table 1 of the revised paper.

---

> ### Author Response · Authors · 2024-11-23
> **Response to Reviewer cjt7 (Part 2)**
>
> **Response to Question 3)**
> During the inference step, frequency information is sampled from the training dataset. To achieve this, we employ kernel density estimation, a nonparametric approach, to estimate the distribution by analyzing the spectral density of the training data. This estimated distribution provides the necessary frequency information for inference, ensuring consistency with the characteristics of the training dataset.
>
> **Response to Question 4)**
> In response to the reviewer's comments, we performed an ablation study on the hyperparameter gamma. Specifically, we conducted experiments varying gamma from 0.8 (the original setting) down to 0.1, focusing particularly on the fMRI dataset, which exhibits a more evenly distributed power spectrum compared to other datasets. The results, presented below, demonstrate that the best performance is achieved with gamma set to 0.8. This indicates that the model performs optimally when the power spectrum reflects an 8:2 ratio, that is, when 80% of the information is concentrated in low frequencies. This has been included in Appendix B.1 of the revised paper.
> | γ   | Context-FID Score ↓ | Correlation Score ↓ | Discriminative Score ↓ | Predictive Score ↓ |
> |-----|----------------------|---------------------|-------------------------|---------------------|
> | 0.8 | **0.071±.076**       | **0.628±.695**      | **0.083±.077**          | **0.066±.032**      |
> | 0.6 | 0.148±.004           | 1.316±.017          | 0.149±.021              | 0.102±.000          |
> | 0.4 | 0.193±.008           | 1.303±.038          | 0.108±.048              | 0.102±.000          |
> | 0.3 | 0.177±.008           | 1.288±.028          | 0.133±.057              | 0.101±.000          |
> | 0.2 | 0.161±.011           | 1.330±.034          | 0.132±.061              | 0.102±.000          |
> | 0.1 | 0.154±.014           | 1.312±.042          | 0.125±.048              | 0.102±.000          |
>
> **Response to Question 5)**
> The self-attention mechanism of the Transformer inherently supports the modeling of long sequences, particularly excelling at incorporating information from previous stages due to its autoregressive nature. This makes it well-suited for capturing complex structures in the denoising process of the diffusion-based model in the time series domain. Furthermore, in the proposed model, the combination of frequency information within the Transformer decoder is achieved through cross-attention. As illustrated in Figure 3, each frequency information is integrated into the time domain representation in the Transformer decoder, allowing for denoising to be carried out. Consequently, the proposed model adopts a decoder structure, rather than an encoder-only configuration, to facilitate this process.
>
> **Response to Question 6)**
> The term "original" refers to the performance obtained when real, rather than synthetic, data is used for training while calculating the predictive score. This metric evaluates the predictive characteristics inherent in the real dataset. The proposed method demonstrates predictive scores comparable to the original on most datasets, with significant improvement observed in the fMRI dataset due to its unique characteristics. Unlike other datasets, the fMRI dataset has higher dimensionality and is particularly noisy. It also focuses on a more specific and limited frequency band, and frequency features are more essential since they reflect neurophysiological information [3]. Consequently, the RNN model used to measure the original performance may have struggled due to these factors. In contrast, the proposed method, leveraging inductive bias through frequency-based learning, exhibits greater robustness to noise, resulting in improved performance on the fMRI dataset.
>
> **Reference**
> ###
>
> [1] Yuan and Qiao, “Diffusion-TS: Interpretable diffusion for general time series generation,” ICLR, 2024.
>
> [2] Carabbe et al., “Time series diffusion in the frequency domain,” ICML, 2024.
>
> [3] Yuen et al., “Intrinsic Frequencies of the Resting-State fMRI Signal: The Frequency Dependence of Functional Connectivity and the Effect of Mode Mixing,” Front. Neurosci., 2019.

---

### Official Review · Reviewer_mVNz · 2024-11-05

**Soundness:** 2
**Presentation:** 2
**Contribution:** 2
**Rating:** 5
**Confidence:** 5

**Summary:**

This paper introduces a novel diffusion model for time series generation that combines both time-domain and frequency-domain information. The model leverages frequency as prior knowledge, allowing it to capture both global and local patterns more effectively during the diffusion process. A key contribution is the introduction of a signal energy-based adaptive frequency selection module, which separates low and high frequency components based on spectral density to better represent temporal features. While the experiments conducted on several public datasets indicate promising results in tasks such as long-range data generation, imputation, and forecasting, the experimental validation could benefit from further solidification across more diverse datasets and conditions.

**Strengths:**

The paper presents a novel approach by integrating frequency-domain information as prior knowledge into a diffusion model for time series generation. This is an innovative idea that addresses limitations in capturing both global trends and local details within time series data. A key strength lies in the signal energy-based adaptive frequency selection module, which intelligently separates low-frequency components for global trends and high-frequency components for semantic details, based on spectral density. This thoughtful and systematic design enhances the model's ability to effectively balance time-domain and frequency-domain information during the denoising process, making the approach both conceptually strong and technically well-crafted. The careful decomposition and integration of frequency components reflect a deep understanding of the challenges in time series generation, providing a promising direction for future research in this field.

**Weaknesses:**

1.	Unclear Problem Definition:
The paper does not clearly define the inputs and outputs for each task, such as prediction and anomaly detection, which makes it difficult for readers to understand the exact setup and application of the model across different tasks. For instance, the specific requirements and challenges for each task are not sufficiently explained, leaving readers unclear on how the proposed approach addresses them.
2.	Choice of Datasets in Generation and Prediction Tasks:
In the generation task, the selection of datasets such as Sines and MuJoCo is problematic, as some evaluation metrics on these datasets yield extremely low values (e.g., 0.001), which may undermine the reliability and interpretability of the results. Additionally, for the prediction task, only two datasets are used, which is too limited to adequately assess the model’s predictive capabilities across different settings.
3.	Outdated Baselines:
Some of the baselines used in the main experiments, such as TimeGAN and TimeVAE, are relatively outdated and may not represent the current state of the field. These older models are less competitive compared to more recent approaches, which diminishes the persuasive power of the experimental comparisons. Including stronger, more recent baselines could significantly enhance the validity and rigor of the results.
4.	Ablation Study
Although the paper performs a thorough ablation study, some components appear to have minimal impact on performance (as shown in Table 3). For example, removing certain components (e.g., high-frequency or adaptive frequency modules) does not substantially degrade the results, raising questions about their necessity. A more targeted ablation analysis focusing on the essential components would make the findings more concise and impactful, avoiding the impression of redundant complexity.

**Questions:**

1.	Could you provide a clearer definition of the tasks, specifying the input and output for each (e.g., anomaly detection, prediction)?
The current description of tasks lacks clarity on the exact inputs and outputs, making it difficult to understand the model's specific application to each task. For instance, a more explicit definition of input-output pairs for prediction and anomaly detection would be helpful.
2.	Could you comment on the choice of datasets for the generation and prediction tasks?
In the generation task, certain datasets like Sines and MuJoCo yield extremely low evaluation metric values (e.g., 0.001), which may undermine the reliability of the results. For the prediction task, only two datasets were used, which may not be sufficient to comprehensively evaluate the model’s predictive capabilities. Expanding on the dataset choices would improve confidence in the findings.
3.	Why were certain outdated baselines chosen for the main experiments, and could you provide comparisons with more recent methods?
Some baselines, such as TimeGAN and TimeVAE, are relatively outdated and may not provide a rigorous benchmark. Including more recent models could provide a stronger validation of the proposed method’s performance and strengthen the comparisons.
4.	In the ablation study, could you elaborate on the necessity of each component tested?
Certain ablation components, such as “w/o high frequency” and “w/o adaptive,” appear to have minimal impact on the results. A more detailed justification for including these components, or focusing only on the critical ones, would make the ablation study more concise and impactful.

---

> ### Author Response · Authors · 2024-11-22
> **Response to Reviewer mVNz (Part 1)**
>
> We appreciate the reviewer’s valuable comments. We address individual concerns below.
>
> **Response to Question 1)**
> Our methodology can be broadly categorized into two key tasks: time-series data generation and task-specific generation, which includes imputation and forecasting. Both tasks follow the learning process illustrated in Figure 3, where the distribution of time-series data is modeled with consideration of frequency information. During training, complete data without missing values is input, sampled through a diffusion process, and output.
>
> For task-specific generation, the conditional distribution is approximately sampled using the pre-trained diffusion model and the gradient of the classifier, following the sampling method in Diffusion-TS [1]. However, frequency information, one of central to our method, is not directly accessible during inference. This limitation is particularly pronounced in data with missing values, where acquisition distortions prevent obtaining normal frequency information. To address this, we utilize the training data set used for pre-training to compute the spectral density through a non-parametric method, kernel density estimation, and use this frequency information during inference. In task-specific generation, the process involves inputting data with missing values alongside the frequency information extracted during training into the approximate sampling method of the conditional distribution via the pre-trained diffusion model. The output is a complete dataset with the missing values imputed or future values predicted, ensuring continuity and coherence in the data. To clarify, we added an explanation of tasks in Appendix B of the revised paper.
>
> **Response to Question 2)**
> We selected datasets with diverse characteristics—such as periodicity, discontinuity, noise levels, regularity of time steps, and correlations across time and features—to evaluate the proposed model, rather than limiting the evaluation to a specific domain or distinct dataset. Regarding the context-FID score (noted by the reviewer for its low numerical value of 0.001), this metric is less sensitive to subtle differences in the generated data distribution as it relies on low-dimensional encoded representations derived from embedded features. Consequently, the low numerical value can be interpreted as a reflection of this property. In this study, we evaluated performance using four metrics, including context-FID, which collectively provide a robust demonstration of the model's reliability. This robustness is further supported by the model's reasonable performance across diverse domains—such as energy (Energy and ETTh), finance (stocks), and healthcare (fMRI)—all of which exhibit complex, real-world patterns.
>
> Our primary focus was on data generation for the main experiments, which is why we initially utilized only two datasets as baseline settings for the forecasting task. In line with the reviewer’s suggestion, we further verified the effectiveness and robustness of the proposed model by incorporating a new solar dataset [2] for prediction tasks.
> | Model | MSE |
> |--------|--------|
> | GP-copula | 9.8e2$\pm$5.2e1 |
> | TransMAF | 9.3e2 |
> | TLAE | 6.8e2$\pm$7.5e1 |
> | CSDI | 9.0e2$\pm$6.1e1 |
> | SSSD | 5.03e2$\pm$1.06e1 |
> | Diffusion-TS | 3.75e2$\pm$3.6e1 |
> | Ours | **3.41e2$\pm$1.4e1** |
>
> Additionally, we extended the imputation experiments [3] on the MuJoCo dataset to include various missing value scenarios, thereby enhancing the reliability and comprehensiveness of our evaluation. The results of the additional experiments have been added in Appendix B.2 of the revised paper.
> | Model        | 70% Missing | 80% Missing | 90% Missing |
> |--------------|-------------|-------------|-------------|
> | RNN GRU-D    | 11.34       | 14.21       | 19.68       |
> | ODE-RNN      | 9.86        | 12.09       | 16.47       |
> | NeuralCDE    | 8.35        | 10.71       | 13.52       |
> | Latent-ODE   | 3.00        | 2.95        | 3.60        |
> | NAOMI        | 1.46        | 2.32        | 4.42        |
> | NRTSI        | 0.63        | 1.22        | 4.06        |
> | CSDI         | **0.24(3)** | 0.61(10)    | 4.84(2)     |
> | SSSD         | 0.59(8)     | 1.00(5)     | 1.90(3)     |
> | Diffusion-TS | 0.37(3)     | 0.43(3)     | 0.73(12)    |
> | Ours         | 0.31(4.5)   | **0.35(5)** | **0.45(1.6)** |
>
> **Response to Question 3)**
> TimeGAN [4] and TimeVAE [5] serve as representative models for GAN-based and VAE-based time series data generation, respectively, with evaluation methods such as predictive and discriminative metrics having been proposed for them. These models were selected for comparison to demonstrate the effectiveness of our approach. Furthermore, we validated the proposed method by comparing its performance with that of more recent models, such as Diffusion-TS [1] and DiffTime [6], offering a comprehensive contrast against both established and cutting-edge techniques.

---

> ### Author Response · Authors · 2024-11-22
> **Response to Reviewer mVNz (Part 2)**
>
> **Response to Question 4)**
> Through ablation studies, we demonstrated that incorporating frequency information significantly enhances performance in the generation task. Moreover, adaptively separating low and high-frequency components, rather than relying solely on specific frequencies, proved to be more effective. Specifically, low-frequency components contribute to capturing global trends, improving overall prediction capabilities, while high-frequency components provide semantic details, enhancing fine-grained generation. This adaptive separation aligns with the inductive bias of the diffusion process, facilitating better data synthesis. Experimental results supported this claim: the predictive score was lower when low-frequency information was excluded, and the discriminative score suffered without high-frequency information due to the absence of semantic priors. Although experiments were conducted with a 24-window length, which had minimal impact on outcomes, the consistent results across trials reinforce our conclusions. To clarify, we added an explanation of the ablation study in Appendix B of the revised paper.
>
> **Reference**
>
> [1] Yuan and Qiao, “Diffusion-TS: Interpretable diffusion for general time series generation,” ICLR, 2024.
>
> [2] Alexandrov et al., “Gluonts: Probabilistic and neural time series modeling in python,” JMLR, 2020.
>
> [3] Alcaraz and Nils, “Diffusion-based time series imputation and forecasting with structured state space models,” arXiv preprint arXiv:2208.09399, 2022.
>
> [4] Yoon et al., “Time-series generative adversarial networks,” NeurIPS, 2019.
>
> [5] Desai et al., “TimeVAE: A variational auto-encoder for multivariate time series generation,” arXiv preprint arXiv:2111.08095, 2021.
>
> [6] Coletta et al., “On the constrained time-series generation problem,” NeurIPS, 2024

---

> > ### Comment · Reviewer_mVNz · 2024-11-25
> > **Response**
> >
> > Thanks for the author to answer my question.

---

> > > ### Comment · Reviewer_mVNz · 2024-12-03
> > > **Response about Frequency-based Condition**
> > >
> > > Could you concisely elaborate on the significant differences introduced by leveraging frequency-domain information as a condition, compared to the conditioning approaches in prior works?

---

> ### Author Response · Authors · 2024-12-03
> **Response**
>
> Recent advancements in time series data generation models increasingly leverage not only the intrinsic information within the data but also additional contextual or metadata-based information [1, 2, 3]. These approaches are designed to incorporate specific constraints (e.g., trends, fixed values) [1] or to account for the unique heterogeneity of the data by utilizing metadata [2]. In particular, FIDE [3] addresses the limitations of time series generation under block maxima by conditioning the generation process on the Generalized Extreme Value (GEV) distribution. However, such methods do not enhance the core capabilities of the diffusion process itself, as they lack a robust conditioning mechanism for modeling under frequency information.
>
> In contrast, our approach achieves improved synthesis by adaptively separating high- and low-frequency components of the data and denoising each component individually. This separation enables the model to better capture the distinct characteristics of the data. Specifically, the effective modeling of low-frequency components, which encode global characteristics, leads to superior predictive scores, particularly excelling in long-term generation tasks. Furthermore, our method demonstrates significant improvements over other baselines in forecasting tasks. Simultaneously, the use of high-frequency semantic information results in strong performance on discriminative scores and exceptional results in imputation tasks, showcasing the versatility and robustness of our approach.
>
> **Reference**
>
> [1] Coletta et al., "On the constrained time-series generation problem," NeurIPS, 2023.
>
> [2] Narasimhan et al., “Time weaver: A conditional time series generation model,” ICML, 2024
>
> [3] Galib et al., “FIDE: Frequency-inflated conditional diffusion model for extreme-aware time series generation,” NeurIPS, 2024

---

### Meta-Review · Area_Chair_EViJ · 2024-12-20

**Metareview:**

This paper proposes a diffusion model for time series generation that integrates both time-domain and frequency-domain information. The paper is novel in combining the information in the frequency-domain and the time domain. However, as raised by the reviewer, there appears to be some deficiency in the experiment design.

**Additional Comments On Reviewer Discussion:**

Although the paper has some merits, such as its innovative approach to combining frequency-domain information as prior knowledge and the signal energy-based adaptive frequency selection module, the issues raised by the reviews are critical. For instance, there appears to be an information leakage in the model design (8Qz6), the experimental settings lack clear definitions of inputs and outputs for different tasks (mVNz), and the choice of baselines is outdated and insufficient for proper comparison (mVNz, cjt7). Although the authors present promising results on several datasets, the paper still needs a major revision before it can be accepted. The decision is reject.

---

### Decision · Program_Chairs · 2025-01-22

Reject